# Coating Thickness Estimation Using a CNN-Enhanced Ultrasound Echo-Based Deconvolution

**DOI:** 10.3390/s25196234

**Published:** 2025-10-08

**Authors:** Marina Perez-Diego, Upeksha Chathurani Thibbotuwa, Ainhoa Cortés, Andoni Irizar

**Affiliations:** 1CEIT-Basque Research and Technology Alliance (BRTA), Manuel Lardizabal 15, 20018 Donostia-San Sebastián, Spain; uthibbotuwa@ceit.es (U.C.T.); airizar@ceit.es (A.I.); 2Department of Electrical and Electronic Engineering, Tecnun, Universidad de Navarra, Manuel Lardizabal 13, 20018 Donostia-San Sebastián, Spain

**Keywords:** coating thickness estimation, deconvolution modelling, CNN, ultrasound pulse-echo

## Abstract

Coating degradation monitoring is increasingly important in offshore industries, where protective layers ensure corrosion prevention and structural integrity. In this context, coating thickness estimation provides critical information. The ultrasound pulse-echo technique is widely used for non-destructive testing (NDT), but closely spaced acoustic interfaces often produce overlapping echoes, which complicates detection and accurate isolation of each layer’s thickness. In this study, analysis of the pulse-echo signal from a coated sample has shown that the front-coating reflection affects each main backwall echo differently; by comparing two consecutive backwall echoes, we can cancel the acquisition system’s impulse response and isolate the propagation path-related information between the echoes. This work introduces an ultrasound echo-based methodology for estimating coating thickness by first obtaining the impulse response of the test medium (reflectivity sequence) through a deconvolution model, developed using two consecutive backwall echoes. This is followed by an enhanced detection of coating layer thickness in the reflectivity function using a 1D convolutional neural network (1D-CNN) trained with synthetic signals obtained from finite-difference time-domain (FDTD) simulations with k-Wave MATLAB toolbox (v1.4.0). The proposed approach estimates the front-side coating thickness in steel samples coated on both sides, with coating layers ranging from 60μm to 740μm applied over 5 mm substrates and under varying coating and steel properties. The minimum detectable thickness corresponds to approximately λ/5 for an 8 MHz ultrasonic transducer. On synthetic signals, where the true coating thickness and speed of sound are known, the model achieves an accuracy of approximately 8μm. These findings highlight the strong potential of the model for reliably monitoring relative thickness changes across a wide range of coatings in real samples.

## 1. Introduction

The rapid expansion of offshore wind energy, which foresees an increase in capacity of 380 GW over the next ten years [1], has intensified the demand for reliable corrosion protection systems capable of withstanding harsh marine environments. Protective coatings applied to steel turbine structures serve as essential barriers against corrosion; however, they are subject to accelerated degradation from prolonged exposure to ultraviolet (UV) radiation, saltwater immersion, and mechanical stress in the marine environment. As a result, the offshore energy sector is increasingly prioritising coating integrity monitoring as an emerging research focus, particularly for detecting early-stage degradation or failure where the coating remains visibly intact but has lost its protective functionality. Even when coatings are well adhered to the substrate, environmental exposure can gradually compromise their effectiveness [2,3].

Monitoring coating degradation is important across various industries, particularly those exposed to harsh and demanding environments. This challenge becomes even more complex when coatings are applied to hard-to-access locations, such as offshore structures, where continuous and reliable monitoring is critical. Current non-destructive testing (NDT) techniques such as ultrasonic, eddy current, magnetic, and optical methods are widely used for coating thickness measurement in industrial settings [4,5,6]. These established methods reliably assess whether a protective coating meets design specifications in terms of uniformity and applied thickness. However, monitoring coating degradation over time presents a more complex challenge. Early-stage degradation often involves subtle changes in material properties, surface conditions, or barrier integrity that may not immediately affect thickness or be detectable by conventional means. In particular, changes in coating properties caused by exposure to harsh environments—for example, marine or industrial conditions—can alter acoustic properties, including the speed of sound, or lead to partial breakdown of the coating. These changes can be detected indirectly through thickness-related measurements. Tracking such relative changes in coating thickness and acoustic response over time can therefore provide valuable insights into the onset and progression of degradation. However, this line of research is still emerging, and more field-relevant experimental data are needed to understand how coatings physically evolve in challenging environments and how these changes affect the assessment of material properties via ultrasonic or other NDT methods. To address this, more advanced and sensitive approaches are needed, which are capable of observing subtle relative changes in signals over time, where recurring or evolving patterns can reveal the earliest indications of coating degradation.

This paper presents our work investigating the potential of ultrasound-based non-destructive testing (NDT) to address the challenge of coating thickness detection for monitoring purposes by offering a reliable and non-invasive means of evaluating the condition of protective coatings and the underlying steel. The motivation for our study is to estimate the thickness of the coating layer applied on the opposite side of the sensor in a three-layer structure, where a steel substrate is coated on both sides. The proposed method uses an ultrasound pulse-echo approach that first obtains the reflectivity sequence of the test medium through a deconvolution model developed from two consecutive backwall echoes, followed by enhanced detection of the coating layer thickness in the reflectivity sequence using a 1D convolutional neural network (1D-CNN).

## 2. State of the Art

This section reviews state-of-the-art developments and ongoing research aimed at improving the detection and monitoring of coating deterioration over time. While not all studies focus specifically on coating degradation, deconvolution model approaches have been widely explored in the literature, and more recently, deep learning (DL) frameworks have emerged as promising tools to address the problem of overlapping echoes in ultrasonic NDT.

Traditional deconvolution methods, including least squares, Wiener filtering, and minimum variance deconvolution [7], rely on prior knowledge of the second-order statistics of both the noise and the input signal. Wiener deconvolution is one of the methods commonly used in the literature for analysing ultrasound backscattered signals, particularly to extract the reflectivity sequence that reveals material flaws. Different works were found in the literature that have applied this convolution-based system modelling and Wiener deconvolution in the scope of ultrasound NDT for materials. For example, Wiener filter-based approaches have been applied for detecting crack-like defects, as discussed in [8,9]. Following Wiener filtering, autoregressive spectral extrapolation was applied to extend the system response beyond the inherent bandwidth limitations of piezoelectric sensors to improve the sparse resolution further. The results of these experiments demonstrated the effectiveness of this combined approach in enhancing the interpretability of ultrasonic signals, allowing features that would otherwise be difficult to discern in the unprocessed data to be more clearly resolved.

Acoustic noise resulting from scattering by the grains within the propagation medium does not have readily known statistics, and conventional Wiener deconvolution, which is based on second-order statistics, could struggle to accurately estimate the flaw impulse response under poor signal-to-noise ratio (SNR) conditions. Addressing this limitation, the authors of [7] proposed a deconvolution technique based on higher-order statistics, which has proven to be more effective in such low SNR environments. This ultrasonic model was applied for defect identification using both synthesised ultrasonic signals and real signals obtained from artificial defects.

As computational resources become increasingly accessible, CNNs and similar architectures are expected to play a significant role in advancing ultrasonic signal interpretation. To address the challenge of overlapping ultrasonic echoes, several recent studies have explored deep learning (DL) frameworks, aiming to perform deconvolution of overlapping echoes through data-driven learning. However, to the best of our knowledge, the integration of explicit deconvolution models with CNNs remains unexplored in the existing literature.

One of the key aspects of a CNN is the training process. To enable accurate learning of a CNN, a large and varied training dataset is essential. This dataset is often generated through finite element (FE) simulations [10,11,12,13] that replicate a realistic propagation of ultrasonic waves in a given medium. Simulated A-scans are augmented with realistic experimental noise, extracted from initial calibration scans, to ensure robustness against variability in measurement conditions. Another common way to generate synthetic A-scan signals is to use sparse spike trains [14], where reflections from different interfaces are modelled as a sequence of discrete impulse events. These signals are then convolved with representative pulse-echo wavelets and added noise conditions approximating real material inspection conditions. To achieve more realistic conditions in synthetic signals, amplitude attenuation, echo dispersion, and phase conversion are incorporated into the modelling process.

A few recent studies have explored the use of convolutional neural networks (CNNs) as end-to-end models for deconvolution in ultrasonic signal processing. In [13], authors proposed to improve axial resolution in ultrasonic testing by using a CNN to separate overlapping echoes, allowing for more accurate estimation of time-of-flight (ToF) and amplitude. The framework was validated experimentally to detect flat bottom holes in an aluminum block with a minimum depth corresponding to λ/4 at 2.25 MHz. A CNN-based approach was proposed in [14] for detecting overlapping ultrasonic echoes trained on simulated data derived from a physics-based echo model. Experimental validation on both simulated datasets and physical phantom samples demonstrated that the proposed network outperforms traditional sparse approximation algorithms, providing significantly improved resolution in layer thickness estimations and exhibiting promising results across various echo overlap scenarios. Notably, the method was demonstrated to detect layer thicknesses as thin as 0.1 mm. In [12], a CNN-based approach was introduced for embedded crack detection within a material. Validation experiments on physical specimens, using A-scan measurements, demonstrated prediction accuracy with average errors of 5.7%, 5.6%, and 8.4% for crack length, location, and orientation, respectively. The evaluated crack sizes, primarily characterised by their length, ranged from 1 mm to 5 mm, representing typical penny-shaped cracks. Sendra et al. (2025) [11] introduced a novel hybrid architecture combining convolutional neural networks (CNNs) with Transformer neural networks (TNNs) for ultrasonic signal deconvolution. The CNN-TNN architecture outperformed the CNN by a 1.81% success rate and exceeded thresholding methods by 17.5% for detecting minimal depths at 0.5 λ, corresponding to resolutions of approximately 1.32 mm (2.25 MHz) and 0.59 mm (5 MHz) on experimental datasets.

A key objective in the proposed methodology is to eliminate the dependency on a reference signal in the deconvolution process—typically required in conventional approaches such as Wiener filter approach for enhancing the sparse resolution of ultrasound responses. To this end, we propose a novel deconvolution model that operates on two consecutive backwall echoes to extract the reflectivity function of the tested medium. The outputs of this model are then processed with a 1D convolutional neural network (1D-CNN) to detect closely spaced reflected impulses within the reflectivity function.

The CNN follows the input–output design proposed in [14], where a one-dimensional input vector encoding overlap information is mapped to an output vector of the same size with smoothed spikes, although with key changes in the synthetic signal generation and with the filters of the architecture adapted to capture the local overlap patterns across the wide range of coating thicknesses in our dataset. In the reference method, synthetic signals consist of a single main backwall echo that sums the contributions from internal layers. They were generated as spike trains with echoes spaced to produce varying degrees of overlap and amplitudes drawn from a controlled range, including both well-separated and closely overlapping reflections, with additive Gaussian noise to achieve a target SNR. However, the methodology proposed by Shpigler et al. [14] showed limited performance across the wide coating thickness range considered in our study (60μm–740μm). Furthermore, the synthetic signals generated based on their approach did not provide a good representation of the signals measured on real samples—a limitation successfully addressed in our method using the k-Wave MATLAB toolbox (v1.4.0). On top of that, the synthetic signals generated in our method explicitly contain multiple echoes within the considered sampling window and simulate acoustic wave propagation through three-layer media. Amplitude attenuation is considered by accounting for reflection, transmission, and material-dependent attenuation coefficients in each layer. These multiple backwall echoes, closely resembling real echoes, are important because the CNN is trained to predict the impulses in the reflectivity function derived from two consecutive echoes rather than directly from time-domain signals with overlapping echoes.

Furthermore, the proposed deconvolution model isolates the reflectivity sequence of the tested medium corresponding to the coating layer of interest, improving the detection of reflections with enhanced sparse resolution and without relying on a reference signal. Building on this, we propose a hybrid framework that integrates the deconvolution model with a CNN for coating thickness estimation.

## 3. Proposed Modelling Approach

In the pulse-echo mode of ultrasonic testing, the reflected signal y(t) can be modelled in the time domain as a convolution of the measurement system impulse response function to an electrical excitation x(t) (assuming the excitation approximates a delta function), the medium’s impulse response h(t), and with additive noise n(t) [7,15,16]. This can be expressed as follows:(1)y(t)=x(t)∗h(t)+n(t)

Deconvolution can then be employed to reverse the effect of convolution and obtain an estimate of the impulse response of the medium, h(t). This impulse response h(t) is a function of time, that represents the characteristics related to the reflections that occur as the sound propagates through the test material in the propagation direction. We refer to this as the reflectivity sequence. If the medium is made up of homogeneous, lossless layers, h(t) will be a sum of impulses, with each impulse’s amplitude determined by the relative change in acoustic impedance between adjacent layers [16]. The underlying assumption is that the measured pulse-echo signal results from the linear convolution of the reflectivity sequence of the material interfaces with the ultrasonic system’s response. Once we model the system response, considering both the transducer characteristics and wave propagation, the deconvolution operation can be applied to reverse the effects of convolution and extract the reflection impulse response from the material interfaces.

In the ultrasound responses from a multilayer material, strong echoes primarily occur at interfaces with significant acoustic impedance differences—the greater the difference, the stronger the reflection. In our study, we focus on steel coated on both sides, as in offshore wind turbine towers, where the relevant interfaces are steel–coating and coating–air. Unlike a bare steel sample, which produces distinct echoes from the steel–air interface, coated samples generate more complex backwall echoes due to overlapping reflections within the layered structure. The degree of overlap depends on coating thickness, sound speed in the coatings, reflection amplitudes, and ultrasonic pulse characteristics such as duration and frequency. Since our goal is to estimate the coating thickness, we apply deconvolution to resolve the partially overlapping steel–coating and coating–air reflections within the main backwall echoes. To do this, we model two consecutive backscattered echoes using a convolution approach (Equation (Equation 1)), assuming the measurement system with a single longitudinal transducer is linear and time-invariant. The model establishes a relationship between the two consecutive echoes by incorporating the impulse responses of each echo’s propagation path and the measurement system, treating successive echoes as delayed and attenuated versions of the transmitted signal. We then apply deconvolution to recover the reflectivity function. To our knowledge, this approach has not been previously applied to model two consecutive backwall echoes and estimate the medium’s impulse response without a known reference signal, forming the basis of our deconvolution model for coating thickness estimation.

Our experience in ultrasonic testing shows that the measured ultrasonic response can be affected by sensor positioning and contact quality. Factors such as couplant thickness, local surface roughness, and coating application variations can affect the signal. Therefore, each sensor–object contact is treated as an independent measurement channel, and deconvolution is performed on this channel basis without relying on reference signals or baselines from other samples. Therefore, any advanced processing applied to achieve our objectives should account for these channel-specific variations, as reflected in the model we present.

### 3.1. Model of the Reflection Response of the Medium

To establish a theoretical basis for analysing back-scattered echoes from a highly reverberant discrete structure, we first studied the simplest case: a single-layer medium representing a bare steel sample. Figure 1 illustrates the reverberation process, showing incident, transmitted, and reflected beams over time. The model assumes a linear, time-invariant system and a flawless, homogeneous material.

A simplified breakdown of echo generation is as follows:Incident Wave: The transducer emits an ultrasonic pulse into the material.First Echo ($ec_{1}$): The pulse reaches the back surface of the material, where a greater portion of the energy is reflected back (due to impedance difference), creating the first echo.Subsequent Echoes ($ec_{2}$): The remaining energy of the original pulse continues to propagate through the material. With each round trip (transmit–receive cycle), additional portions of energy are reflected at the back surface, resulting in successive echoes.

In the ultrasound response from a bare steel sample, successive echoes result from multiple reflections at the same interface (steel backwall–air). Each echo represents a round trip of the ultrasonic wave between the transducer and the backwall. Two consecutive received echoes can be expressed using a convolution formulation as follows.$$\mathrm{First}\mathrm{Echo}:\;ec_{1}\left(t\right)=x\left(t\right)*h_{ec_{1}}\left(t\right)+n_{1}\left(t\right)$$

Here, $ec_{1}\left(t\right)$ represents the first backwall echo reflected in the received signal as a convolution of the measurement system impulse response x(t) with the impulse response of the total propagation path for the first echo $h_{ec_{1}}\left(t\right)$, with $n_{1}\left(t\right)$ accounting for additive noise.

Assuming that the second echo is simply a further propagated and attenuated version of the transmitted signal, and considering x(t), the measurement system impulse response, $h_{ec_{1}}\left(t\right)$, the impulse response of the total propagation path for the first echo, and $h_{ec_{2}}\left(t\right)$ as the propagation path traversed by the first echo to produce the second echo, we can model the second echo as follows:$$\mathrm{Second}\mathrm{Echo}:\;ec_{2}\left(t\right)=x\left(t\right)*h_{ec_{1}}\left(t\right)*h_{ec_{2}}\left(t\right)+n_{2}\left(t\right)$$
where $ec_{2}\left(t\right)$ represents the second backwall echo as a convolution involving both $h_{ec_{1}}\left(t\right)$ and an additional term for its own propagation path $h_{ec_{2}}\left(t\right)$ with additive noise $n_{2}\left(t\right)$.

Converting these representations into the frequency domain yields:



$$EC_{1}\left(f\right)=X\left(f\right)\times H_{ec_{1}}\left(f\right)+N_{1}\left(f\right)$$


$$EC_{2}\left(f\right)=X\left(f\right)\times H_{ec_{1}}\left(f\right)\times H_{ec_{2}}\left(f\right)+N_{2}\left(f\right)$$



Here, we assume that the signal has high SNR, meaning that the noise terms $N_{1}\left(f\right)$ and $N_{2}\left(f\right)$ are relatively small compared to the signal components. Therefore, they are not explicitly carried forward in the next step. Assuming our signal has high SNR, we can approximate the relationship between two consecutive echoes as(2)$$EC_{2}\left(f\right)=EC_{1}\left(f\right)\times H_{ec_{2}}\left(f\right)$$

This provides a direct relationship between the first and second echoes, with $H_{ec_{2}}\left(f\right)$ representing the additional propagation path that the second echo travels relative to the first echo, and this leads us to:(3)$$H_{ec_{2}}\left(f\right)=\frac{EC_{2}\left(f\right)\times EC_{1}^{*}\left(f\right)}{|EC_{1}{\left(f\right)|}^{2}+K}$$

Here, * denotes the complex conjugate and *K* is a small regularisation term introduced to stabilise the deconvolution process, which implicitly accounts for noise effects. Its purpose is to prevent division by very small values of $|EC_{1}{\left(f\right)|}^{2}$, which could otherwise amplify noise and lead to unstable estimates of $H_{ec_{2}}\left(f\right)$. The exact value of *K* is usually chosen experimentally, based on the noise level present in the measurements.

The assumptions we made during this modelling are as follows:The transducer is inherently band-limited.Any type of noise present in the system is assumed to be additive.The incident wave is modelled as a longitudinal plane wave.The measurement system is assumed to be linear and time-invariant (LTI); therefore, deconvolution techniques can be applied.

To obtain $H_{ec_{2}}\left(t\right)$ in the time domain, an inverse fast Fourier transform (IFFT) is performed:$$h_{ec_{2}}=\mathrm{IFFT}\left(H_{ec_{2}}\right)$$ For the bare steel sample, since all echoes propagate solely through the steel, the term $h_{ec_{2}}\left(t\right)$ corresponds to the medium’s reflectivity function.

Next, we extend the analysis to a three-layer medium, with a coating layer on each side of the steel substrate, under the condition that the steel thickness is much greater than that of the coatings and therefore the main consecutive backwall echoes can be clearly separated. Figure 2 shows the energy reflection paths for steel samples coated on both sides, where the first and third layers are coatings. The focus is on isolating the propagation paths of the first two main backwall echoes. It is important to note that reverberations within coating layers are not considered in this figure.

For the coating media, $ec_{1}$ and $ec_{2}$ represent the main backwall echoes, each a composite signal formed by multiple reflections within the layered media. These reflections partially overlap, producing a single, complex event in the received signal. So as to account for transmission and reflection at the boundaries of the multilayered structure, two semi-infinite layers (couplant/transducer and air) are also defined:Layer 0:  Couplant/TransducerLayer 1:  Rear coatingLayer 2:  SteelLayer 3:  Front coatingLayer 4:  Air

The reflections are caused by impedance mismatches at the interfaces, where the impedance of the medium depends on the density and speed of sound:(4)$$Z_{k}=\rho_{k}c_{k}$$
where

$\rho_{k}$: material’s density [kg/m^3^] in medium *k*;

$c_{k}$: speed of sound [m/s] in medium *k*.

Reflection and transmission coefficients at normal incidence, when going from medium *k* to medium k+1, define how the amplitude of the stress or pressure of the elastic wave changes [17]:(5)$$\begin{array}{cc} r_{k,k+1} & =\frac{Z_{k+1}-Z_{k}}{Z_{k+1}+Z_{k}}, \end{array}$$(6)$$\begin{array}{cc} t_{k,k+1} & =\frac{2Z_{k+1}}{Z_{k+1}+Z_{k}}. \end{array}$$

The reflection coefficient $r_{k,k+1}$ is always less than 1 in magnitude, while the transmission coefficient $t_{k,k+1}$ can exceed 1 when $Z_{k+1}>Z_{k}$, corresponding to an increase in the stress or pressure amplitude at the interface. Despite this, the transmitted energy remains physically consistent.

The relevant approximate values of these coefficients [18,19] for the three-layer system are listed in Table 1. These coefficients were calculated using relevant acoustic impedance values of each layer.

To understand the contribution of the ultrasound pulse propagation path to the consecutive echoes measured by the transducer at the top of the sample, we approximate the impulse response of each layer as an ideal delta, assuming linearity and neglecting any possible dispersion:$$p_{k}=m_{k}\;\delta \left(t-t_{k}\right),$$
with $m_{k}$ being the mean attenuation in the *k*-th layer due to propagation, and $t_{k}$ the time it takes for the ultrasound pulse to traverse the layer.

Furthermore, at each interface between layers, the pulse amplitude is modified according to the corresponding reflection or transmission coefficient of Equations (Equation 5) and (Equation 6), respectively.

As illustrated in Figure 2, when the ultrasound pulse passes through the layered medium for the first time, the first backwall echo can be approximated as the convolution of the $p_{k}$ responses of all layers traversed by the ultrasound pulse, scaled by the corresponding reflection and transmission coefficients (refer to Equations (Equation 5) and (Equation 6)) at each layer interface:$$\begin{array}{cc} ec_{1} & =t_{01}\;t_{12}\;r_{23}\;t_{21}\;t_{10}\;p_{1}*p_{2}*p_{2}*p_{1}+t_{01}\;t_{12}\;t_{23}\;r_{34}\;t_{32}\;t_{21}\;t_{10}\;p_{1}*p_{2}*p_{3}*p_{3}*p_{2}*\;p_{1}\\ \; & =t_{01}\;t_{10}\;t_{12}\;t_{21}p_{1}*p_{1}*p_{2}*p_{2}*\left(r_{23}\delta \left(t\right)+t_{23}\;r_{34}\;t_{32}\;p_{3}*p_{3}\right) \end{array}$$

Approximate values of reflection and transmission coefficients for the acoustic interfaces in the layer medium are provided in Table 1. Each contribution to the first backwall echo can be approximated as the product of the reflection and transmission coefficients along the corresponding propagation path. Using the reflection and transmission coefficient values provided in Table 1, the two main paths for the first echo have approximate total coefficients:$$\begin{array}{c} |t_{01}\;t_{12}\;r_{23}\;t_{21}\;t_{10}|\approx 0.266,\\ |t_{01}\;t_{12}\;t_{23}\;r_{34}\;t_{32}\;t_{21}\;t_{10}|\approx 0.105. \end{array}$$

These values indicate the relative contribution of the main propagation paths to the first backwall echo in the model. The first path, with a total coefficient of approximately 0.266, is the dominant contributor to the echo, while the second path, with a coefficient of approximately 0.105, contributes less. This indicates that most of the energy in the first backwall echo originates from the reflection at the steel–front coating interface.

The impedance mismatch at the steel–rear coating interface is larger than at the rear coating–couplant/transducer interface. Therefore, it can be assumed that most of the energy in the second echo comes from the reflection of the elastic wave at the steel–rear coating interface, characterised by the reflection coefficient $r_{21}$ (before receiving the first echo), which then propagates again through the layered medium. Consequently, the second echo can be approximated as:$$\begin{array}{cc} ec_{2} & =t_{01}\;t_{12}\;p_{1}*p_{2}*p_{2}*\left(r_{23}\delta \left(t\right)+t_{23}\;r_{34}\;t_{32}\;p_{3}*p_{3}\right)\\ \; & *\left(r_{21}\;r_{23}\;t_{21}\;t_{10}\;p_{2}*p_{2}*p_{1}+r_{21}\;t_{23}\;r_{34}\;t_{32}\;t_{21}\;t_{10}\;p_{2}*p_{3}*p_{3}*p_{2}*p_{1}\right) \end{array}$$

As we can see, the second echo arises from multiple propagation paths, each involving a different sequence of reflections and transmissions. To compare their relative contributions, calculated total coefficients for each path are:$$\begin{array}{cc} C_{1} & =|t_{01}\;t_{12}\;r_{23}\;r_{21}\;r_{23}\;t_{21}\;t_{10}|\;=\;0.183,\\ C_{2} & =|t_{01}\;t_{12}\;r_{23}\;r_{21}\;t_{23}\;r_{34}\;t_{32}\;t_{21}\;t_{10}|\;=\;0.072,\\ C_{3} & =|t_{01}\;t_{12}\;t_{23}\;r_{34}\;t_{32}\;r_{21}\;r_{23}\;t_{21}\;t_{10}|\;=\;0.072,\\ C_{4} & =|t_{01}\;t_{12}\;t_{23}\;r_{34}\;t_{32}\;r_{21}\;t_{23}\;r_{34}\;t_{32}\;t_{21}\;t_{10}|\;=\;0.028. \end{array}$$
The dominant contributor to the second echo is the path associated with $C_{1}$, which corresponds to the reflection from the steel–front coating interface without any prior propagation through the coating. The two paths associated with $C_{2}$ and $C_{3}$ involve propagation through the front coating and arrive at the transducer at the same time, so their contributions add constructively, enhancing the overall reflection from the front coating.

In the expression obtained for $ec_{2}$, by factoring out the common coefficients and convolving impulses, we obtain:$$\begin{array}{cc} ec_{2} & =t_{01}\;t_{10}\;t_{12}\;t_{21}\;r_{21}\;p_{1}*p_{1}*p_{2}*p_{2}*p_{2}*\;p_{2}\\ \; & *\left(r_{23}\delta \left(t\right)+t_{23}\;r_{34}\;t_{32}\;p_{3}*p_{3}\right)*\left(r_{23}\delta \left(t\right)+t_{23}\;r_{34}\;t_{32}\;p_{3}*p_{3}\right)\\ \; & =ec_{1}*r_{21}\;p_{2}*p_{2}*\left(r_{23}\delta \left(t\right)+t_{23}\;r_{34}\;t_{32}\;p_{3}*p_{3}\right)\\ \; & =ec_{1}*h_{ec_{2}} \end{array}$$

Therefore, considering that $h_{ec_{2}}$ corresponds to the impulse response of the propagation path associated only with the second echo, the second echo can be expressed as the convolution of the first echo with $h_{ec_{2}}$:$$ec_{2}=ec_{1}*h_{ec_{2}}$$
where $h_{ec_{2}}$ is:$$\begin{array}{cc} h_{ec_{2}} & =r_{21}\;p_{2}*p_{2}*\left(r_{23}\delta \left(t\right)+t_{23}\;r_{34}\;t_{32}\;p_{3}*p_{3}\right). \end{array}$$

It is important to note that $h_{ec_{2}}$, which explicitly represents the propagation path specific to the second echo, can be expressed in terms of two consecutive echoes, echo 1 and echo 2. The same approach can be applied to other pairs of consecutive echoes, such as the second and third backwall echoes, to recover similar reflection information. Thus, $h_{ec_{2}}$ can be generalised to *h*, defined for any selected pair of consecutive echoes, where echo 1 refers to the first arriving echo and echo 2 refers to the second arriving echo. It contains the main reflection impulses, including reflections from the steel–front coating interface and from the front coating–air interface. Substituting the definition of $p_{k}$ and applying delta function properties yields the following expression for *h*:(7)$$\begin{array}{cc} h\equiv h_{ec_{2}} & =r_{21}\;r_{23}\;m_{2}^{2}\left[\delta \left(t-2t_{2}\right)+m_{3}^{2}\;\frac{t_{23}\;r_{34}\;t_{32}}{r_{23}}\;\delta \left(t-2t_{2}-2t_{3}\right)\right]. \end{array}$$

The resulting impulse response is the sum of two positive spikes, with the second attenuated and time-shifted. This arises because $r_{23}$ and $r_{34}$ are both negative (as $Z_{2}>Z_{3}>Z_{4}$), $r_{21}$ is also negative ($Z_{1}<Z_{2}$), and the transmission coefficients are always positive. Here, we assume that the coating is a single layer with homogeneous material properties. The two spikes are separated by a time interval of $2t_{3}$, which corresponds to the time-of-flight of the ultrasound pulse within the front coating layer:$$2t_{3}={\mathrm{ToF}}_{3}=\frac{2d_{3}}{c_{3}},$$
where $c_{3}$ is the speed of sound in the front coating. Therefore, by knowing $c_{3}$, the thickness $d_{3}$ of this layer can be inferred from the impulse response of the test medium.

Thus, considering this analysis, even for a three-layer medium representing steel coated on both sides, the relation given by Equation (Equation 2) remains valid. Hence, the model we obtained for bare steel (refer to Equation (Equation 3)) can be applied in this case too.(8)$$H\left(f\right)=\frac{EC_{2}\left(f\right)\times EC_{1}^{*}\left(f\right)}{|EC_{1}{\left(f\right)|}^{2}+K}$$

In characterising the term *K* in our model, we adopt the concept of regularisation, which is commonly used in Wiener filter-based deconvolution applications. This approach is typically used to minimise the least mean square error (MSE) between the estimated signal and the true signal, making it particularly effective for extracting impulse responses while mitigating noise effects. In our case, it was adapted to optimise the deconvolution process between the two consecutive echoes by providing a balance between noise suppression and signal preservation. Hence, *K* is calculated by the power spectral density of the first backwall echo ($EC_{1}$) to determine the noise term as given by Equation (Equation 9).(9)$$K=10^{-2}\times {\left|EC_{1}\left(f\right)\right|}_{max}^{2}$$

Additionally, this approach is driven by a key aspect of our model: processing data based on individual contact-based channels, each accounting for its unique characteristics. Given that consecutive backwall echoes are obtained within a short time frame, we make the reasonable assumption that the measurement system remains almost consistent for both echoes. The primary difference between these echoes lies in their propagation paths and any additive noise. Under these conditions, using $EC_{1}$ to calculate the noise factor provides an estimate of the noise floor that reflects the actual measurement environment common to both echoes.

## 4. Methodology

The proposed methodology involves several steps. Starting from an echo signal, the deconvolution model is applied to obtain the reflectivity function of the medium. Synthetic signals are generated to train a 1D-CNN, which is then applied to real reflectivity functions to detect the impulses and estimate the coating thicknesses. The overall process is illustrated in Figure 3.

The development of the deconvolution model is described in Section 3.1. In this section, we present the measurement setup, the characteristics of the coated samples used in our test cases, the generation of synthetic signals, and the implementation of the deconvolution and CNN models.

### 4.1. Measurement Setup and Materials

For the measurement setup, an ultrasound testbed (UTB) system (see Figure 4) was used, controlled by a PC or laptop running MATLAB to initiate the ultrasound measurements and collect the corresponding responses [20]. The UTB system generates longitudinal ultrasound waves, and the responses are recorded based on the pulse-echo method.

The V111 probe from Olympus [21] was used as the ultrasound probe for measurements. The selected V111 transducer from Olympus has a centre frequency of 8.07 MHz and a −6 dB fractional bandwidth of 81.86%. Furthermore, the ADC used in the UTB samples at $f_{s}=125$ MHz, corresponding to 8 ns per discrete sample of the digitised signal.

Table 2 lists the samples used for the analysis. Here, we have used samples that have a coating thickness varying from 60 to 740 μm.

### 4.2. Deconvolution Model Implementation

The ultrasound responses from the coating samples are analysed using the deconvolution model developed (described in Section 3.1). The overall digital signal processing workflow is shown in Figure 5. The received ultrasound responses include echoes from various interfaces within the investigated samples, as well as noise, which can be either acoustic or electrical in nature. Acoustic noise arises from backscattering due to grain boundaries in metallic materials, where the boundaries between small crystals scatter ultrasonic waves, resulting in clutter or noise in the received signal. Additionally, electrical noise from the UTB or ultrasound sensor node may also be present in the received measurements. To effectively identify flaw-related information, it is important to enhance the characteristics of the bandpass filter to remove noise while ensuring that signals related to the flaws are preserved. Therefore, at the beginning, the raw ultrasound signals are filtered using the bandpass filter with a passband of 2–15 MHz. Analysis of the frequency spectrum for both steel sample types (structural steel: S355J2G3 and S235JR), including bare and coated samples, using the Olympus 8 MHz V111 sensor, reveals that flaw-related information predominantly resides below 15 MHz. In contrast, higher frequencies exhibit some noise presence. The bandpass filter is a FIR (Finite Impulse Response) filter designed in MATLAB using the fir1 function with a Hanning window, a filter length of 115, and cutoff frequencies of 2–15 MHz, based on a sampling frequency of 125 MHz. The filter coefficients were quantised for implementation in our custom, non-commercial measurement system, UTB. The envelope of the bandpass-filtered signal was then used to window the echoes, following the technique described in our previous work [22]. To obtain the envelope, the bandpass-filtered signal was rectified and subsequently passed through a FIR low-pass filter.

During ultrasonic testing, 25 consecutive measurements are typically performed at each measurement spot. Before processing begins, these measurements are averaged to enhance the results minimising random errors and fluctuations that may occur. To minimise artifacts that can arise during frequency analysis, a Tukey window is applied to the echoes prior to performing the fast Fourier transform (FFT). This step ensures that any discontinuities at the boundaries of the echo windows are smoothed out, enhancing the accuracy of the frequency representation. The Tukey window is defined by a tapering parameter, which in this case is set to 0.15. This parameter determines the proportion of the window that transitions smoothly to zero at its edges, in reducing spectral leakage. A small tapering value is used so that important information from the echoes is preserved while still mitigating edge effects.

After obtaining the reflectivity function in the frequency domain, an inverse fast Fourier transform (IFFT) is performed to obtain the time-domain reflectivity function. The result is then normalised (*h*) to bring it to a common scale. In the recovered reflectivity function, the impulses corresponding to each interface resemble typical ultrasound pulse-echoes. Rather than appearing as ideal delta functions, they represent the convolution of the true reflectivity with the band-limited impulse response of the piezoelectric transducers. Additionally, practical factors such as non-ideal excitation pulses, attenuation, frequency-dependent (dispersion) transducer ringing, reverberations within coatings, and noise can further affect the observed reflectivity.

Then, once we obtain the reflectivity function for a particular measurement from the three-layered medium, we extract the impulses corresponding to the main interfaces of the medium: the steel backwall reflection (steel–front coating interface) and the front coating backwall reflection (front coating–air interface). The time-of-flight (ToF) between these two impulses corresponds to the ultrasound signal propagating within the front coating, which can be used to determine the thickness of this layer if the speed of sound in the material is known. To improve the accuracy of impulse detection in the reflectivity sequence, we employ the CNN trained on a synthetic dataset. This approach improves both detection accuracy and resolution compared to simple thresholding methods applied directly to the reflectivity sequence. The generation of synthetic signals and the CNN architecture are discussed in Section 4.3 and Section 4.4, respectively.

### 4.3. Synthetic Signal Generation

CNN models require a large amount of labelled data for training. However, measurements from our real samples are limited in number and are labelled only with approximate coating thickness values, as the exact thickness and speed of sound are unknown. Therefore, to train the model effectively with signals of predefined ground truth, synthetic signals are generated—some for training and a few for testing—while the ultimate goal is to use the model to predict the coating thickness of the real signals.

Initially, an attempt was made to create synthetic signals by overlapping reference generalised impulses extracted from bare steel measurements, with varying attenuations and separations corresponding to the front coating thickness. This trial followed the approach in [14], but used impulse signals rather than echo signals. However, these synthetic signals differed significantly from the real signals, preventing the CNN from fully capturing some real features. Factors such as dispersion, pulse broadening, reverberations within the coating layers, and oscillations from processing—arising from the deconvolution of adjacent echoes using an approximate noise factor—could not be properly accounted for, and analytically incorporating them to accurately mimic real signals proved challenging.

Therefore, the k-Wave MATLAB toolbox [23] was employed to simulate the propagation of an ultrasound pulse through a layered medium replicating the three-layer sample, with an impedance-matching couplant on one side and air on the other, as illustrated in Figure 2. The k-Wave toolbox is based on the k-space pseudospectral method and enables FDTD simulation of elastic wave propagation in 2D or 3D media [24]. In this case, 2D elastic simulations are employed, as the layers are laterally uniform and only vary in the stacking (depth) direction, which allows for reduced computational cost.

This approach allows the pressure field at the transducer position to be measured during simulation, producing the echo signal, which is then processed using the same steps as for the real signals to generate a synthetic impulse response closely resembling the real ones. The following subsections describe the simulation and processing steps required to obtain the synthetic signals used to train the CNN model, as summarised in the upper part of the diagram in Figure 3.

#### 4.3.1. Definition of the Propagation Medium

For each simulation, the material properties of the layers are defined by randomly sampling values from predefined ranges that reflect realistic values for low carbon steel (S235JR or S355J2G3) and various coating materials. These values are listed in Table 3. The rear and front coating properties are chosen independently. The coatings are modelled as a single layer with uniform properties, even though in practice they may consist of multiple sublayers with different properties. This approximation is justified because the acoustic impedances of sublayers in standard coating systems are generally similar, so that variations among them are minor and hence the dominant reflections occur primarily at the coating–steel and coating–air interfaces. Fixed, known values are used for the steel, couplant, and air, with the latter two modelled as semi-infinite media.

#### 4.3.2. Simulation Setup

The simulation is conducted using a 2D computational grid of size 256×64, with the axial direction (x-axis) aligned with the ultrasound propagation path, making its resolution particularly critical. The physical dimensions of the domain are set to $x_{\mathrm{size}}=6.5\;\mathrm{mm}$ and $y_{\mathrm{size}}=20\;\mathrm{mm}$. To prevent non-physical reflections at the domain boundaries, an anisotropic absorbing boundary layer, implemented as a perfectly matched layer (PML), is applied. The total simulation time is set to 7.5μs, which is sufficient to capture at least three consecutive main echoes. The temporal grid is determined based on the maximum sound velocity in the system, ensuring convergence by satisfying the Courant–Friedrichs–Lewy (CFL) condition. The time step is computed as $\Delta t=0.1\;\cdot \;\Delta x/c_{\max}$, where $c_{\max}=c_{\mathrm{steel}}$ is the highest wave speed in the medium.

For the simulation, the reference echo from a pulse-echo measurement in bare steel (S235JR or S355J2G3) is used directly as the incident excitation pulse, after being oversampled to match the simulation’s time resolution. A 13 mm-wide line source is used to represent the transducer, matching the actual transducer’s diameter. The excitation is introduced as a normal (longitudinal) stress in the $s_{xx}$ direction, with wave propagation along the x-axis. To mimic the focused acoustic beam typically emitted by real transducers, the excitation is modulated in the spatial domain along the source line using a Gaussian function with standard deviation of $\sigma ={\mathrm{radius}}_{\mathrm{transducer}}/2$. The same line also serves as the sensor, where the received signal is computed by spatially averaging the pressure field over the sensor aperture in order to obtain the one-dimensional A-scan. Once the grid, material properties, source, and sensor definitions are established, the simulation is executed to model ultrasound wave propagation and reflection in the multilayered medium.

#### 4.3.3. Synthetic Signals from Simulation

From the simulation, the pressure field is recorded at all sensor positions to obtain a B-scan signal, which is then spatially averaged to produce an A-scan comparable to real measurements. This signal is then downsampled to match the real acquisition sampling rate of 125 MHz and subsequently processed using the same steps applied to the experimental data: band-pass filtering, echo windowing, and deconvolution between adjacent echoes to extract the reflectivity signal. To improve robustness, impulse responses between echo 1 and echo 2, as well as between echo 2 and echo 3, are considered. This choice is based on the assumption that each echo represents a repeated reflection of the previous one through the layered medium. Ideally, it should not matter which pair of echoes is selected. However, in practice, each echo may contain reverberation effects and overlapped reflections that are not consistently preserved throughout propagation, which can influence the resulting impulse response. The final synthetic reflectivity signal, consisting of 512 sample points, is used as the input to the CNN.

#### 4.3.4. Ground Truth Definition

The ground truth is defined as a spike train, or sparse vector, with two non-zero entries corresponding to the reflective interfaces steel–front coating and front coating–air, based on the impulse response of the test medium defined in Equation (Equation 7).

For each simulation, the amplitudes are determined by the reflection, transmission, and attenuation coefficients, computed from the defined material properties. Although the incident US excitation pulse is broadband, the frequency dependent attenuation due to propagation is computed using a fixed central frequency of f=8MHz.

Since the focus is on the relative amplitude and time separation between the two spikes, the common attenuation is omitted, and the common displacement due to the steel layer thickness is treated as a variable reference parameter from the post-processing stage, $t_{2_{ref}}$. Therefore, the ground truth is a normalised and displaced version of the theoretical impulse response:(10)$$\begin{array}{cc} h_{norm} & =\left[\delta \left(t-2t_{2_{ref}}\right)+m_{3}^{2}\;\frac{t_{23}\;r_{34}\;t_{32}}{r_{23}}\;\delta \left(t-2t_{2_{ref}}-2t_{3}\right)\right]. \end{array}$$

The resulting spike train is smoothed by convolving it with a Gaussian kernel. This transforms the sparse vector into a smoother curve, facilitating the CNN’s learning task while preserving the temporal positions of the echoes, as carried out in [14]. Finally, the ground truth signal is downsampled to match the 512-sample length of the synthetic signal, so that the spike separation of the ground truth in samples is given by:(11)$$\begin{array}{c} {\mathrm{ToF}}_{3}\;\left(\mathrm{samples}\right)={\mathrm{ToF}}_{3}\left(t\right)\times f_{s}=2t_{3}\times f_{s}=\frac{2d_{3}}{c_{3}}\times 125\;\mathrm{MHz} \end{array}$$

### 4.4. CNN-Network

A 1D convolutional neural network (CNN) is implemented in Python 3.9.18 using TensorFlow/Keras and following an architecture inspired by [14]. Instead of explicitly modelling the physical system, the network is designed to implicitly learn the reflectivity response of layered media. All operations in the model are purely convolutional, and the parameters are learned through backpropagation. The output of the network is a smooth impulse train representing the location and amplitude of each echo event.

The model is trained to learn the mapping between impulse responses and their corresponding smoothed spike trains, which represent reflection events within a three-layer medium. In this configuration, the rear coating is impedance-matched to the transducer through a couplant. The training dataset consists of synthetic signals generated using k-Wave simulations. Each sample includes an impulse response (i.e., the reflectivity function) and a smoothed spike train used as ground truth.

Signals and targets are flattened, stacked, and reshaped into arrays of shape (N,T,1), where *N* is the number of samples, *T* the number of time steps, and the final dimension represents a single-channel input. The dataset is randomly split into a training set (80%) and a validation set (20%).

The CNN architecture is summarised in Figure 6. It consists of six convolutional layers of 32 filters each, all using ReLU activation, followed by a dropout layer and a final convolutional layer. The network uses wide kernels in the early layers to capture broad temporal features, followed by progressively narrower kernels in the subsequent layers, with dilated convolutions to increase the receptive field without reducing temporal resolution. This design is important because the model must handle a wide variation of coating thicknesses, and the local patterns corresponding to different thicknesses are difficult to capture with a single filter size. A dropout layer with a rate of 0.2 is included to mitigate overfitting, and the final 1×1 convolutional layer maps the features to a one-dimensional smoothed spike train.

The model is compiled using the Adam optimiser and a mean squared error (MSE) loss function, which is also used as a performance metric. Training is performed with a batch size of 100 and a maximum of 30 epochs. Two callback functions are employed: ReduceLROnPlateau, which reduces the learning rate by a factor of 0.2 if the validation loss plateaus for 12 epochs, and EarlyStopping, which halts training after 20 epochs of no improvement in validation loss, restoring the weights from the best-performing epoch.

This architecture effectively captures both the global structure of the signal and localised spike events by combining wide and narrow convolutional receptive fields. The output is a sparse, smooth signal that preserves the temporal structure of reflection events in the simulated media.

## 5. Results and Discussion

The impulse response of the test medium is obtained by post-processing the simulated received pressure signal, which results from propagating the real ultrasound pulse through the material. Therefore, it is important to directly compare both the synthetic, filtered pulse-echo signal and the impulse response with the real ones. Figure 7 shows the synthetic signal that best matches the real one for that coating type, identified by maximising the cross-correlation between the synthetic and real signals, as well as between their corresponding impulse responses. The similarity metric in each subfigure, ranging from 0 to 1 (with values closer to 1 indicating higher similarity), is calculated as the absolute maximum of the cross-correlation between the plotted real and synthetic reflection impulse response signals. For the signals plotted here, this metric ranged between 0.90 and 0.94.

Differences between the simulated and real signals may arise for the following reasons:**Measurement system impulse response and experimental noise.** The non-ideal response of the transducer and measurement system cannot be explicitly modelled in k-Wave. In the simulations, the excitation is defined directly as a normal stress with a temporal waveform taken from the first backwall echo of bare steel measurements (one for each steel type, S235 and S355, randomly assigned in the simulations), and spatially modulated with a Gaussian kernel to mimic a focused aperture. Since this reference signal is obtained from pulse-echo measurements in bare steel performed with the same transducer used for the coated samples, it is assumed to capture the effect of the transducer’s impulse response. However, it is not strictly representative of the actual transmitted pulse, and thus remains a simplification. Experimental noise, which refers to random fluctuations in the measurement system—e.g., electrical noise, electronic jitter, and environmental vibrations—is also present in the real signals and cannot be incorporated into the simulation.**Frequency-dependent attenuation and dispersion.** k-Wave accounts for frequency-dependent attenuation and the corresponding dispersion using a quadratic power-law model ($\alpha \left(f\right)\propto f^{2}$). This corresponds to an ideal thermoviscous loss law, which does not always reflect the actual frequency-dependent absorption of real materials.**Homogeneity and isotropy.** The simulations assume constant and isotropic material properties within each layer. This is a good approximation for steel, and although the coatings are intended to be uniform, their application process may lead to small variations in homogeneity that are not captured in the model.**2D approximation.** A 2D geometry is used primarily for computational efficiency, as full 3D simulations at the required resolution would be prohibitively expensive. The 2D assumption is a good approximation when the out-of-plane dimension is much larger than the propagation path, as in our case, where the sample thickness is small compared to its lateral area. However, 2D simulations cannot capture out-of-plane scattering, diffraction, or mode conversion, and they confine energy differently than in 3D, which may slightly alter attenuation and energy distribution.**Single-layer coating approximation.** In the simulations, the front and rear coatings were modelled as single layers. In contrast, some of the real commercial coatings consist of multiple sublayers. However, the impedance mismatch between these sublayers is assumed to be small, so the dominant contribution to reflections arises from the last sublayer adjacent to air. It is possible that these internal sublayers introduce minor reverberations or phase shifts in the ultrasonic field, but such effects have not been incorporated at this stage.

This gap between synthetic and experimental signals is difficult to avoid, as it arises from unknown or non-ideal aspects of the measurement setup that are challenging to emulate and incorporate into simulations. In spite of this, the synthetic signals are considered to contain oscillatory patterns sufficiently similar to those in the real signals, allowing the CNN to recognise these patterns and generalise effectively to unseen data.

By closely analysing the synthetic signals in Figure 7, it can be observed that the ground truth does not always align with the most prominent peaks due to the close overlap of reflections or their physical characteristics. For very thin coatings, as in Figure 7a, the overlap is so close that the presence of the coating layer can be difficult to discern. Nevertheless, even for these thin coatings, the reflectivity signal is more oscillatory than that of bare steel, indicating a slight but noticeable overlap. In Figure 7b, for example, the overlap causes partial amplitude cancellation: the local maximum nearest the ground truth second spike is smaller than the subsequent oscillation maxima due to the additive nature of the overlapping reflections. Moreover, the limited bandwidth of the transducer and additional reverberations within both coatings create oscillations that make it difficult to extract the actual ground truth visually or with conventional signal processing methods. In contrast, the CNN can learn the local patterns corresponding to the subtle differences in peak amplitude, spacing, and oscillation frequency between the steel backwall and front-coating backwall reflections, enabling the extraction of spikes corresponding to the reflectivity function with higher accuracy.

From now on in the manuscript, results are reported in samples, and for better understanding, the corresponding micrometre values are provided using a reference speed of sound of 2500 m/s, so that 1 sample corresponds to approximately 10 μm, according to Equation (Equation 11).

For model development, the CNN was trained on approximately 10,000 synthetic signals, with 80% of the data used for training and 20% reserved for validation. After thirty epochs, the model reached a validation mean squared error (MSE) of $5.61\times 10^{-5}$. The model was first evaluated on predicting the ground truth of 100 unseen synthetic signals, for which the exact thickness and speed of sound of the front coating are known. Some of the predictions are plotted in Figure 8. The ToF, corresponding to the separation between the two main reflection peaks, was computed by performing a parabolic interpolation around each maximum of the smoothed spike train to estimate their true locations, and then calculating the distance between them.

The main peak prediction is very well aligned with the ground truth, while the second peak contributes more significantly to the prediction error of the ToF. To illustrate this, the second peak prediction is zoomed in the subfigures. We observe that the model has learned to identify local patterns and avoids detecting the reflection spike corresponding to the coating at local maxima locations when this is not defined in the ground truth. Conversely, when the ground truth defines a reflection at those positions, the model is able to detect it correctly. For example, in Figure 8a, where the overlap is particularly close, the second ground-truth spike does not align with any local maxima of the synthetic signal. Nevertheless, the model predicts it accurately, thanks to training on synthetic signals with similar features and a ground truth capturing the overlap.

Table 4 summarises the statistical metrics of the ToF prediction errors for the 100 unseen synthetic signals, including absolute mean error (MAE), median, standard deviation (of the MAE), root mean square error (RMSE), minimum, maximum, skewness, and kurtosis. The metrics are computed from the absolute errors, as the distribution of errors shows no systematic positive or negative bias. The positive skewness indicates that most of the prediction errors are concentrated toward the lower end of the range, while the few larger errors appear as outliers. The kurtosis above three suggests a peaked distribution, meaning that the majority of errors are clustered near the median (0.62 samples), and extreme deviations toward the maximum are relatively rare. The model accuracy is reported as RMSE (0.8 samples), which provides a single metric that accounts for both the magnitude and variability of the prediction errors, corresponding to an equivalent thickness error of approximately 8μm (assuming a front-coating speed of sound of 2500m/s).

Also, for a clearer visualisation of the prediction errors, the synthetic signals were grouped according to their coating thickness into defined intervals, and the mean absolute errors for each interval are plotted in Figure 9, with the number of signals in each interval indicated above the bars. The results show that the errors are consistent across intervals, reflecting a systematic predictive behaviour across different thicknesses. Only in the penultimate interval is a slightly higher mean error and standard deviation observed; however, since this analysis is based on a set of 100 synthetic signals with 10–15 signals per interval, this variation lies within the expected range.

The model’s prediction on synthetic signals constitutes the best-case scenario, as these synthetic signals—although unseen during training—originate from the same simulation process as the training data, unlike the real signals, which are acquired through laboratory measurements. After this evaluation, the model was applied to estimate the unknown front coating thicknesses in nearly 100 real signals.

In the case of real signals, additional small spikes are sometimes predicted, originated from minor reflections that the model cannot consolidate into just two main spikes. However, two main maxima are predicted in all cases. As with the synthetic signal predictions, to compute ${\mathrm{ToF}}_{3}$, a parabolic interpolation was applied to the two maxima to refine their locations, and the distance between these points was calculated. This provides an estimated value with subsample resolution, overcoming the limitation imposed by the 125MHz sampling frequency of the real measurement system.

Figure 10 shows the predictions made by the network for one sample from each set, including the ground truth spike train and the ${\mathrm{ToF}}_{3}$ derived from the spike separation. Correspondingly, Table 5 shows the mean predicted front coating thicknesses for all the samples from the different sets. In the laboratory, measurements were performed at different positions on each sample, since the coating thickness may vary along the surface due to non-uniform application. The results are therefore grouped by sample position, with each set–position group including 5 to 11 measurements performed on the same area (top or bottom) of each sample. The values in the table are given in samples and cannot be directly converted to thickness, since the exact speed of sound in the real coatings is unknown. However, assuming a reference coating speed of sound of 2500 m/s, an equivalence of 1 sample ≈10μm has been applied for comparison with the nominal thickness values provided by the manufacturers. Taking into account that the actual conversion from samples to thickness depends on the coating’s speed of sound, and that the nominal thickness values themselves exhibit some tolerance due to the coating application process, the predictions are well-aligned with expectations. Nonetheless, the actual prediction error cannot be computed as was done for the synthetic signals, because the true thickness of each real sample and the speed of sound in its coating are not precisely known. In some cases, variation in thickness across measurements at the same position of a sample is observed. This was previously noted in the real signals, where differences between reflections in the impulse responses were visually noticeable. Such variation occurs because the signals were collected after measuring the same area multiple times, and the measurement spot may not have been exactly identical each time, meaning that the actual front coating thickness could genuinely differ. Therefore, these variations cannot be attributed to the prediction model’s precision.

The model successfully estimates the coating thicknesses within the expected value range, demonstrating good generalisation to unseen data. This also indicates that the synthetic signals are sufficiently representative of the real signals for accurately predicting the two main spikes from which the ${\mathrm{ToF}}_{3}$ can be extracted. The maximum tested coating thickness is approximately 740μm, while the minimum is around 60μm. Given the transducer’s central frequency of ∼8 MHz and the reference coating speed of sound 2500 m/s, this minimum corresponds to a detectable thickness slightly below λ/5. To further validate the model’s performance on real signals, it would be valuable to test samples with accurately calibrated thicknesses and known sound speeds for each layer.

## 6. Conclusions and Future Work

In this paper, a novel CNN-enhanced ultrasound echo-based deconvolution model was developed to estimate the front-side coating thickness in three-layer coated steel samples. By deconvolving two consecutive backwall echoes, the method removes the dependency on the incident ultrasound pulse and provides an estimated reflectivity function that highlights the two main interfaces of interest: the steel–front coating interface and the front coating–air interface. To generate signals with known coating properties and to train the network, 2D k-Wave elastic simulations were used to create synthetic signals. Finally, a 1D-CNN was designed to discriminate the reflections of the steel backwall and the front-coating backwall within the impulse response of the test medium.

The model is able to estimate coating thicknesses between 60 µm and 740 µm, with a minimum detectable thickness of about λ/5 for an 8 MHz transducer—surpassing resolution levels typically reported in the literature for this frequency. While the exact performance on real signals cannot be precisely quantified due to unknown coating properties, results on synthetic signals demonstrate a thickness prediction accuracy of ±0.8 samples (≈8μm). Overall, these findings highlight the potential of the model to generalise to experimental data and to serve as a reliable tool for monitoring relative changes in coating thickness.

The current model has been validated on 5 mm thick steel, but its principle is expected to extend to thicker substrates, provided sufficient transducer energy compensates for material absorption. Future work could also incorporate multilayer coatings in simulations to account for additional reflections between sublayers, as well as explore different coating properties to enhance the model’s applicability. Moreover, while all signals in this study were obtained with an 8 MHz transducer, extending the model to different centre frequencies and frequency bands would enhance generalisation. Using higher-frequency transducers would improve axial resolution in the micrometre range but reduce penetration depth due to higher material absorption, potentially diminishing the contribution of the coating to the received pulse-echo signal. In addition, the regularisation term in the deconvolution model could be further refined in future work to better suppress the ringing artefacts around the reflectivity impulses in the reflectivity function. This would be particularly beneficial for improving the reliability of detecting impulses in the reflectivity function for thinner coatings, or when reflections from individual sublayers become somewhat significant in the main ultrasound response.

## Figures and Tables

**Figure 1 sensors-25-06234-f001:**
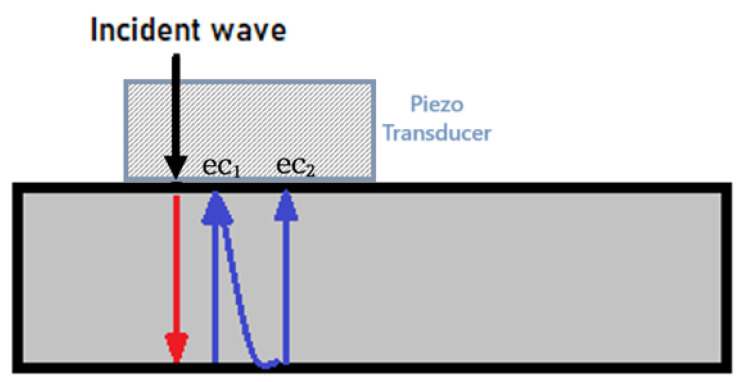
Ultrasound reflection profile for bare steel.

**Figure 2 sensors-25-06234-f002:**
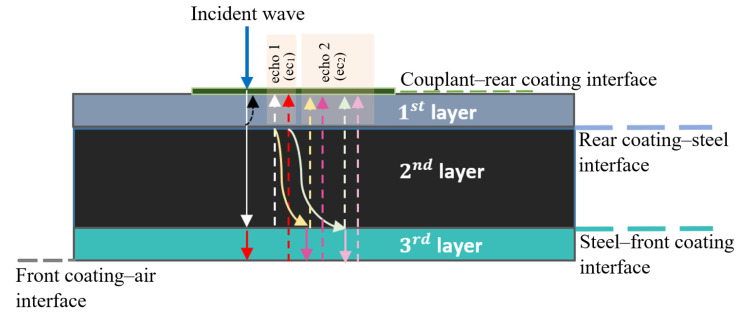
First two reflecting echoes in a three-layer medium: representing a 5 mm steel sample coated on both sides (the reverberation effects within coating layers are neglected).

**Figure 3 sensors-25-06234-f003:**
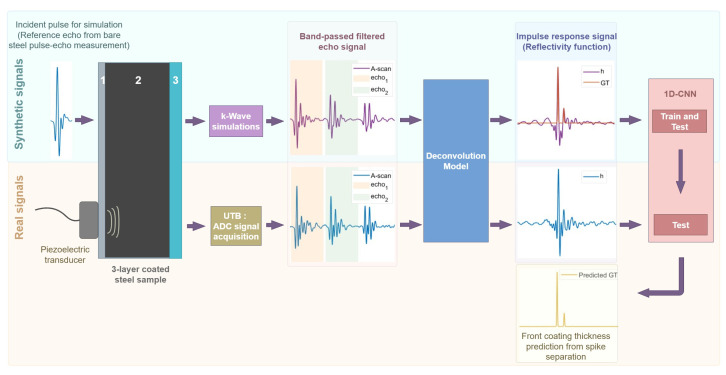
Proposed approach combining deconvolution and 1D-CNN for coating thickness estimation.

**Figure 4 sensors-25-06234-f004:**
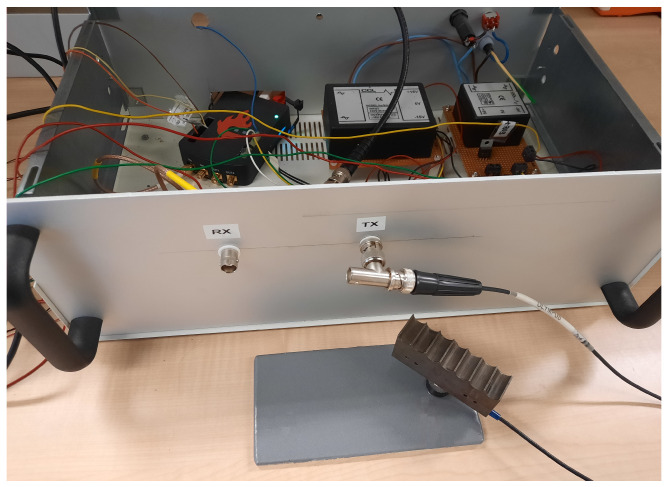
Ultrasound testbed (UTB) measuring a coated sample.

**Figure 5 sensors-25-06234-f005:**
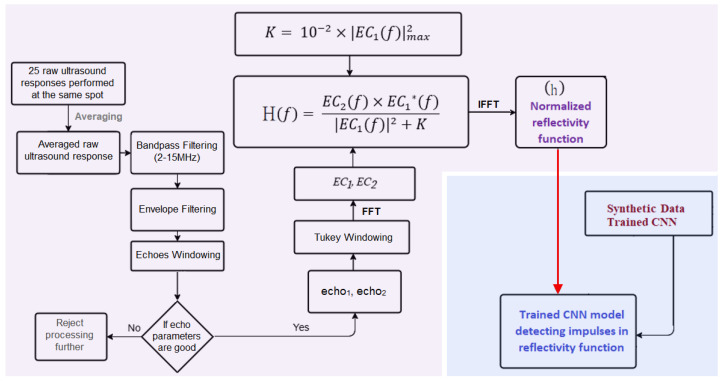
Block diagram of the process for obtaining the reflectivity function and detecting impulses within it. Here, * represents the complex conjugate.

**Figure 6 sensors-25-06234-f006:**
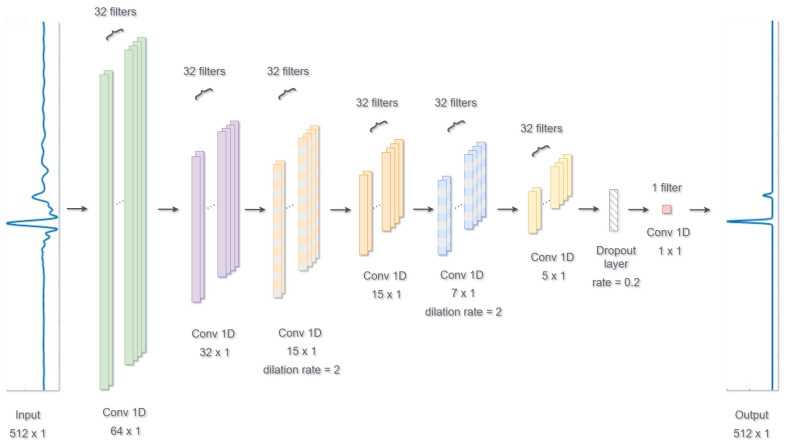
CNN architecture used for smoothed spike reconstruction.

**Figure 7 sensors-25-06234-f007:**
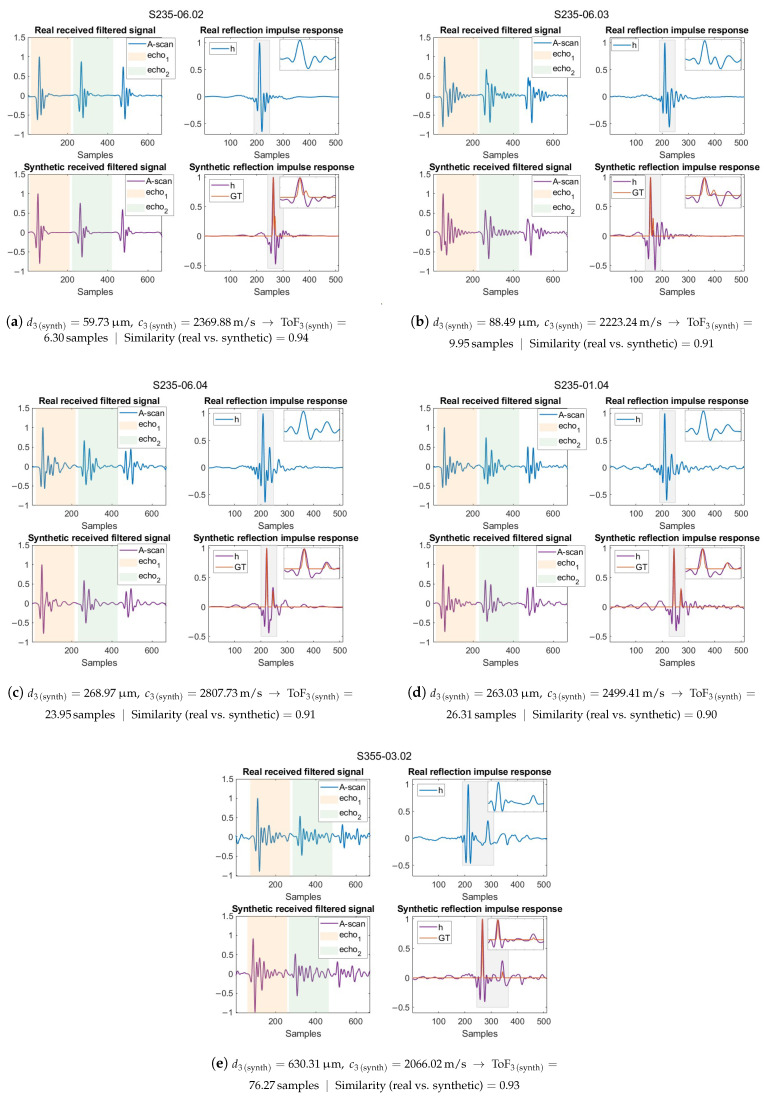
Graphical comparison between the real and synthetic echo signals and impulse responses.

**Figure 8 sensors-25-06234-f008:**
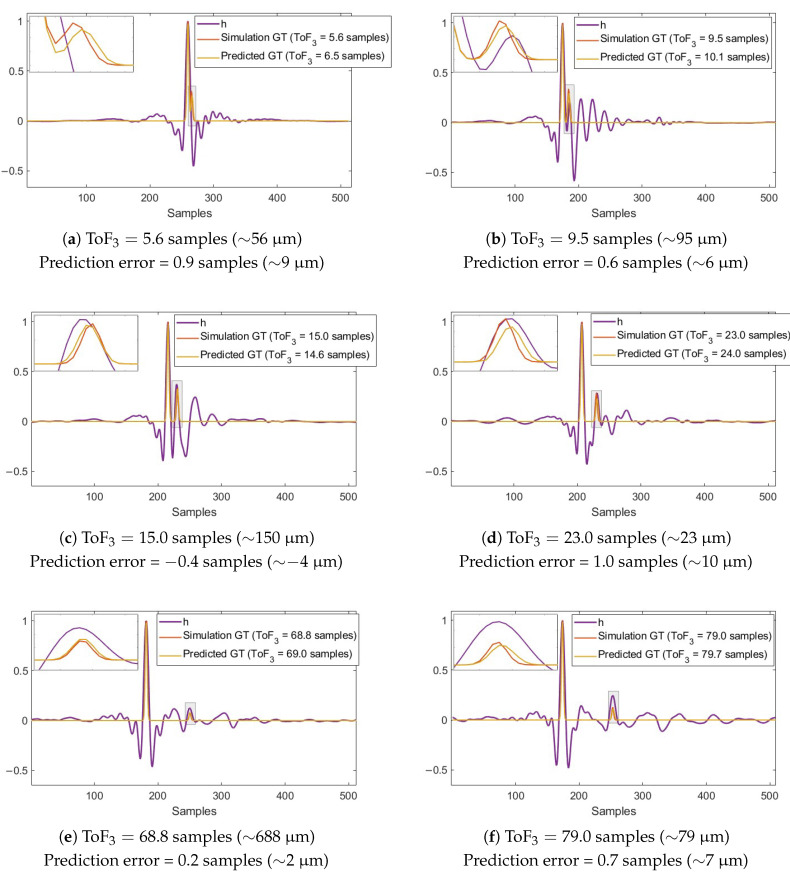
Ground truth prediction of some synthetic reflectivity signals.

**Figure 9 sensors-25-06234-f009:**
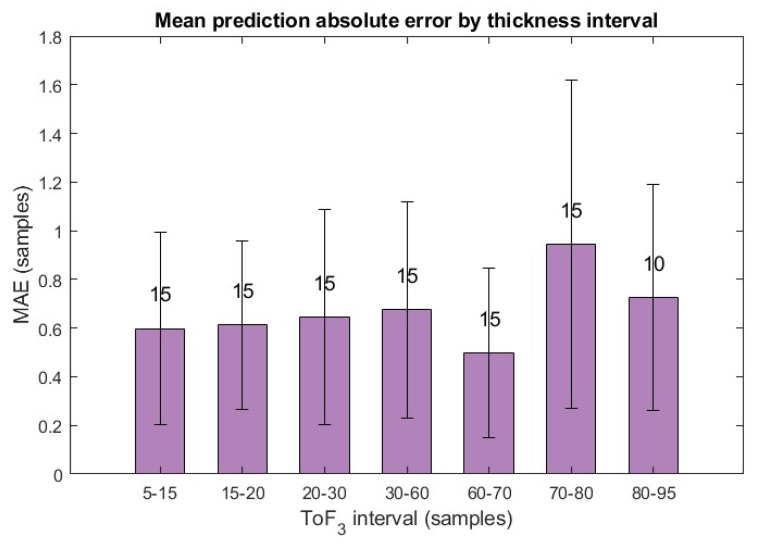
Bar plot of the mean absolute prediction error of synthetic signals, grouped by coating thickness intervals. The number above each bar indicates the number of signals in that interval.

**Figure 10 sensors-25-06234-f010:**
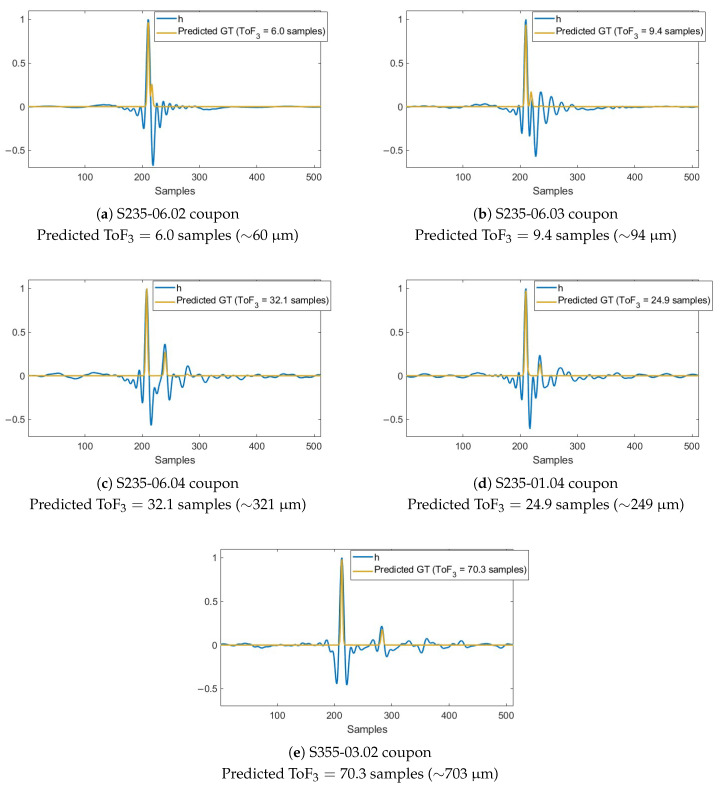
Ground truth prediction of real reflectivity signals from each sample type.

**Table 1 sensors-25-06234-t001:** Approximate reflection ($r_{k,k+1}$) and transmission ($t_{k,k+1}$) coefficients for interfaces between adjacent layers. Only the coefficients of layers that interact in the three-layered system have been considered.

**Reflection** $\left[r_{ij}\right]$** and Transmission** $\left[t_{ij}\right]$						
	**k+1 layer**	0: Couplant/ Transducer	1: Rear Coating	2: Steel	3: Front Coating	4: Air
k **layer**						
0: Couplant/ Transducer			$r_{01}=0.16$ $t_{01}=1.16$			
1: Rear Coating		$r_{10}=-0.16$ $t_{10}=0.84$		$r_{12}=0.83$ $t_{12}=1.83$		
2: Steel			$r_{21}=-0.83$ $t_{21}=0.17$		$r_{23}=-0.83$ $t_{23}=0.17$	
3: Front Coating				$r_{32}=0.83$ $t_{32}=1.83$		$r_{34}=-0.999$ $t_{34}=0.001$

**Table 2 sensors-25-06234-t002:** Test samples with applied coating thickness.

Sample ID	Coating Thickness	
	Rear Side	Front Side
01.04	Metallic 1 (∼36 μm)	Multilayer: commercial product Sigmacover 1 + Sigmadur 2 (∼300 μm)
06.02	Metallic 1 (∼41 μm)	Metallic 1 (∼60 μm)
06.03	Metallic 1 (∼41 μm)	Metallic 1 (∼80 μm)
06.04	Metallic 1 (∼26 μm)	3-layer coating: dimetcote 9 + sigmafast 278 + sigmadur 550 (∼278 μm)
03.02	1–2 coats epoxy (∼468–500 μm)	Zn-rich primer + epoxy intermediate coat+PU coat (∼630–740 μm)

***Note:*** Here, the sample’s “rear side” means the measurements performed side of the sample. The “front side” refers to the exterior coating. Metallic 1 is a hybrid anticorrosion coating that combines silicon alkoxides with zinc compounds. Blue colour represents samples of S355J2G3 steel, and pink represents samples of S235JR structural steel.

**Table 3 sensors-25-06234-t003:** Material property ranges used in the simulation.

Material	Thickness [µm]	Compressional Velocity [m/s]	Shear Velocity [m/s]	Density [kg/m^3^]	Compressional Attenuation [dB/(MHz^2^·cm)]	Shear Attenuation [dB/(MHz^2^·cm)]
Couplant	Semi-infinite	1500	–	1100–1400	0	–
Rear Coating	20–50 (S235) 400–500 (S355)	2000–3000	–	1100–2500	0.5–2 (S235) 0.5–2 (S355)	–
Steel	5000	5950	3120	7850	0.03	0.05
Front Coating	50–500 (S235) 600–900 (S355)	2000–3000	–	1100–2500	0.5–2 (S235) 0.5–2 (S355)	–
Air	Semi-infinite	343	–	1.2	0.002	–

**Table 4 sensors-25-06234-t004:** Statistical metrics of the CNN’s ${\mathrm{ToF}}_{3}$ prediction errors (in samples), evaluated on 100 synthetic signals not seen during training.

	MAE	Median	Std. Dev. (MAE)	RMSE	Min	Max	Skewness	Kurtosis
**${\mathrm{ToF}}_{3}$ error (samples)**	0.67	0.62	0.46	0.81 (⇒∼8 μm)	0.02	2.40	0.85	3.84

**Table 5 sensors-25-06234-t005:** Statistical metrics of the CNN’s predicted ${\mathrm{ToF}}_{3}$ values on nearly 100 real signals, grouped by sample ID, test piece, and measurement position. Mean and standard deviation are reported in samples, with the equivalent micrometre values given in parentheses (1sample≈10μm). Nominal coating thickness values are included for reference.

Sample ID	Test Piece	Bottom		Top		Nominal
		Mean (Samples)	Std (Samples)	Mean (Samples)	Std (Samples)	
S235-06.02	021	6.37 (∼64 μm)	0.28 (∼3 μm)	6.65 (∼67 μm)	0.09 (∼1 μm)	60 μm
S235-06.03	031	9.43 (∼94 μm)	0.39 (∼4 μm)	9.34 (∼93 μm)	0.48 (∼5 μm)	80 μm
	039	9.77 (∼98 μm)	0.59 (∼6 μm)	10.71 (∼107 μm)	1.08 (∼11 μm)	
S235-06.04	041	25.62 (∼256 μm)	0.32 (∼3 μm)	32.29 (∼323 μm)	0.80 (∼8 μm)	278 μm
	042	36.59 (∼366 μm)	1.02 (∼10 μm)	34.06 (∼341 μm)	1.59 (∼16 μm)	
S235-01.04	04	25.12 (∼251 μm)	1.97 (∼20 μm)	24.90 (∼249 μm)	1.79 (∼18 μm)	300 μm
S355-03.02	02	70.39 (∼704 μm)	0.60 (∼6 μm)	73.88 (∼739 μm)	1.45 (∼15 μm)	630–740 μm

## Data Availability

The data supporting the conclusions of this article will be made available by the authors on request.

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
