# Peer review of "Coating Thickness Estimation Using a CNN-Enhanced Ultrasound Echo-Based Deconvolution"

_sensors, 2025, doi:10.3390/s25196234_

Round 1

Reviewer 1 Report

Comments and Suggestions for Authors

The application of AI technology and deep learning methods has become increasingly popular in recent years. This paper presents a novel approach by combining convolutional neural networks (CNN) with ultrasonic thickness measurement, representing a meaningful contribution to the field of ultrasonic non-destructive testing and evaluation. The reviewer acknowledges the significance and potential impact of this work. However, several issues need to be addressed to improve the clarity, conciseness, and scientific rigor of the manuscript before it can be considered for publication. The major and minor concerns are outlined below:

1. Redundancy and Structural Issues:
The manuscript currently contains several redundant sections that could be significantly streamlined.

-The background information in the abstract is overly detailed and should be shortened to focus on the core objective, methodology, and key findings.

-The purpose of Section 2 is unclear. Its removal does not seem to compromise the overall integrity of the paper. The authors should either clarify its contribution or consider integrating its content into other sections or removing it altogether.

-The conclusion section is too lengthy and descriptive. It should be revised to clearly highlight the most important findings and implications of the study, avoiding repetition of results.

-Tables 1 and 2 appear unnecessary and do not provide meaningful information to support the main arguments. The authors should justify their inclusion or remove them to improve manuscript conciseness.

2. Figure Organization and Presentation:
The logical flow of figures should be improved to enhance readability and clarity.

-Key figures such as Figure 2, which illustrates the overall methodology and objective (i.e., coating thickness measurement), should be moved earlier in the manuscript. This would help readers immediately understand the motivation and framework of the study before delving into theoretical details.

-Some subfigure labels are missing (e.g., in Figure 8), and others (e.g., Figure 9) would benefit from clearer or revised labeling. The authors should ensure all subfigures are properly labeled and consistently formatted.

3. Technical and Scientific Concerns:

The reviewer is particularly curious about how the method distinguishes between surface and bottom reflection signals when the coating is very thin. In Figures 8 and 9 (the first two subfigures), there is no apparent change in the signal, yet the trained model is able to detect the presence of a coating and predict its thickness accurately. The authors should provide a more detailed explanation of the underlying physical or learned features that enable this discrimination.

Given the model's apparent sensitivity, the reviewer questions its generalizability. How robust is the model when applied to unknown coating types or under different experimental conditions (e.g., different substrates, temperatures, or transducer setups)? The authors should discuss the model’s robustness and potential limitations in real-world or variable environments. If the authors' model can truly distinguish such aliased signals, I believe this would be a remarkable technological achievement. However, the current experiments were conducted only under the authors' own controlled conditions, which are still limited in validating the general effectiveness of the method. 

4. Lack of Comparative Analysis:
The manuscript primarily focuses on the performance of the proposed CNN-based method but lacks comparative studies with conventional or alternative techniques. For instance, could traditional water-immersion ultrasonic testing or other established methods achieve similar accuracy in measuring the same coating thicknesses? The authors are strongly encouraged to include a comparative analysis to better contextualize the advantages and limitations of their approach.

In summary, while the study presents a promising and innovative application of deep learning in ultrasonic NDT, the manuscript requires substantial revision to address the issues of redundancy, figure organization, technical explanation, and comparative validation. Addressing these concerns will significantly strengthen the paper’s scientific value and readability.

Author Response

Comment 1: The background information in the abstract is overly detailed and should be shortened to focus on the core objective, methodology, and key findings.  

Response 1: We have revised the abstract to make it more concise as is suggested, reducing the background details while preserving the technical content. 

Comment 2: The purpose of Section 2 is unclear. Its removal does not seem to compromise the overall integrity of the paper. The authors should either clarify its contribution or consider integrating its content into other sections or removing it altogether. 

Response 2: Thank you for pointing this out. We realized that Section 1 and Section 2 were not clear enough. Therefore, we reduced the content of Section 1, which is the Introduction, focusing on the context and motivation of this research. And in Section 2, we present the relevant references related to our proposal and how we pretend to contribute to the scientific community with our outcomes. 

Comment 3: The conclusion section is too lengthy and descriptive. It should be revised to clearly highlight the most important findings and implications of the study, avoiding repetition of results.  

Response 3: Thank you for your comment. Considering your suggestion, we have tried to be more concise and highlight the most important findings of the proposed study. Non-essential discussion and detailed explanations of results have been moved to the preceding section, while the conclusion now focuses on summarizing the key outcomes and emphasizing directions for future work. 

Comment 4: Tables 1 and 2 appear unnecessary and do not provide meaningful information to support the main arguments. The authors should justify their inclusion or remove them to improve manuscript conciseness.  

Response 4:Thank you for the comment. Tables 1 and 2 were originally included to illustrate the reflection and transmission coefficients for the interfaces considered in our testing of the three-layer object and due to attenuation in the coatings, strong reverberation effects within the coatings are not expected. Their purpose was to provide an overall understanding of the proportion of ultrasound energy reflected or transmitted at each interface. We have decided to retain Table 1 to support the echo modeling with multiple propagation paths. However, following the reviewer’s suggestion,  Table 2 in the first submitted manuscript, titled ‘Table 2. Approximate attenuation factors for each layer at f = 8 MHz, expressed as functions of the layer thickness dk (cm). mk2 (dk ) represents the round-trip attenuation factor’ has been removed. 

We understand that presenting this content under the ‘Theory’ section (Section 2 in the first submission) may have caused some confusion. In response to the reviewers’ collective feedback, we have removed that section, and the essential content for the model has now been integrated into the section “Proposed Modelling Approach”. 

Comment 5: Key figures such as Figure 2, which illustrates the overall methodology and objective (i.e., coating thickness measurement), should be moved earlier in the manuscript. This would help readers immediately understand the motivation and framework of the study before delving into theoretical details.  

Response 5: Thank you for your suggestion. We understand the confusion and with the aim of clarifying this, we have improved the content of Section 1 (Introduction). Specifically, we have explained better the motivation and framework of our scientific contribution at the end of Section 1 whereas in Section 2 we discuss the State of the Art related to this research describing how our approach is beyond the State of the Art. We believe that these improvements enhance the readability and clarity of our motivation and contribution. 

Comment 6: Some subfigure labels are missing (e.g., in Figure 8), and others (e.g., Figure 9) would benefit from clearer or revised labeling. The authors should ensure all subfigures are properly labeled and consistently formatted.  

Response 6: Thank you for your comment. Subfigure labels have been added to Figure 8, including the ToF value for each subfigure and the corresponding prediction error. In Figure 9, the predicted ToF value has also been added in the label, and the steel type has been specified, clarifying that the previous ID referred to a measured coupon. We also have reviewed all other subfigures to ensure that labeling and formatting are consistent throughout. 

Comment 7: The reviewer is particularly curious about how the method distinguishes between surface and bottom reflection signals when the coating is very thin. In Figures 8 and 9 (the first two subfigures), there is no apparent change in the signal, yet the trained model is able to detect the presence of a coating and predict its thickness accurately. The authors should provide a more detailed explanation of the underlying physical or learned features that enable this discrimination.  

Response 7: Thank you for pointing this out. A closer examination of the signals in the first two subfigures of Figures 7, 8, and 9 reveals subtle differences in their oscillatory behavior that allow discrimination between coating reflections. For instance, in the second subfigure of Figures 7, 8, and 9, the small peak following the first reflection indicates a partially destructive overlap, while the subsequent oscillations suggest a slightly larger coating thickness compared to the signal in the first subfigure. These characteristics are consistently captured in the synthetic reflectivity signals, which the model has learned to associate with specific coating thicknesses during training on synthetic data with known ground truth. Essentially, the model leverages these subtle patterns—changes in peak amplitude, spacing, and oscillation frequency—rather than relying solely on the most prominent peaks, to accurately detect the presence of a coating and predict its thickness. A more detailed explanation, with references to the subfigures, has been included in the manuscript at the new line 670:  

“By closely analyzing the synthetic signals, it can be observed that the ground truth does not always align with the most prominent peaks due to the close overlap of reflections or their physical characteristics.  

For very thin coatings, as in Subfigure 7a, the overlap is so close that the presence of the coating layer can be difficult to discern. Nevertheless, even for these thin coatings, the reflectivity signal is more oscillatory than that of bare steel, indicating a slight but noticeable overlap. In Subfigure 7b, for example, the overlap causes partial amplitude cancellation: the local maximum nearest the ground truth second spike is smaller than the subsequent oscillation maxima due to the additive nature of the overlapping reflections. 

Moreover, the limited bandwidth of the transducer and additional reverberations within both coatings create oscillations that make it difficult to extract the actual ground truth visually or with conventional signal processing methods. In contrast, the CNN can learn the local patterns corresponding to the subtle differences in peak amplitude, spacing, and oscillation frequency between the steel backwall and front-coating backwall reflections, enabling the extraction of spikes corresponding to the reflectivity function with higher accuracy.” 

And in the description of Figure 8, at new line 694: 

“The main peak prediction is very well aligned with the ground truth, while the second peak contributes more significantly to the prediction error of the ToF. To illustrate this, the second peak prediction is zoomed in the subfigures. We observe that the model has learned to identify local patterns and avoids detecting the reflection spike corresponding to the coating at local maxima locations when this is not defined in the ground truth. Conversely, when the ground truth defines a reflection at those positions, the model is able to detect it correctly. For example, in Subfigure 8a, where the overlap is particularly close, the second ground-truth spike does not align with any local maxima of the synthetic signal. Nevertheless, the model predicts it accurately, thanks to training on synthetic signals with similar features and a ground truth capturing the overlap.” 

Comment 8: Given the model's apparent sensitivity, the reviewer questions its generalizability. How robust is the model when applied to unknown coating types or under different experimental conditions (e.g., different substrates, temperatures, or transducer setups)? The authors should discuss the model’s robustness and potential limitations in real-world or variable environments. If the authors' model can truly distinguish such aliased signals, I believe this would be a remarkable technological achievement. However, the current experiments were conducted only under the authors' own controlled conditions, which are still limited in validating the general effectiveness of the method.  

Response 8: Our study focuses on coating-thickness monitoring for wind-turbine towers, using an ultrasonic sensor node that is intended to be deployed inside the tower to evaluate the external coating. The ultrasound measurements used in this study were conducted under laboratory conditions with our measurement setup to establish and validate the methodology. We acknowledge the reviewer’s concern about generalizability. 

At this stage, we have not yet performed field trials. But some samples we used to test the model were prepared with standard coating types (e.g., NORSOK 7A and other commercial coatings) commonly used in field applications. In addition, we tested two steel types as substrates. Together, we believe that these tests provide a representative basis for evaluating the reflection energy from coating-substrate interfaces that closely reflect real-application scenarios and allow us to establish a reference for how such reflections behave in this type of configuration. Since reflection energy depends mainly on acoustic impedance contrast (in pulse-echo mode we detect the reflected part of the energy from acoustic interfaces), we expect the approach to be valid for similar coating–steel configurations under our defined measurement setup conditions and testing scenario (three layer medium where the substrate is coated from both sides).    

Further, we note that generalization will mainly depend on factors such as coating composition, substrate properties and thickness, and the transducer setup (center frequency and bandwidth). In this study, standard and commercial coating types were tested, and as long as signal attenuation from the coating layer is not excessive which can occur for certain coatings, such as with zinc-rich layers, the method is expected to remain valid. Full validation under diverse real-world conditions for the purpose of generalization will require future extended testing of different coating properties. And this is foreseen to be done as our future work aligned with this research activity.  

Comment 9: The manuscript primarily focuses on the performance of the proposed CNN-based method but lacks comparative studies with conventional or alternative techniques. For instance, could traditional water-immersion ultrasonic testing or other established methods achieve similar accuracy in measuring the same coating thicknesses? The authors are strongly encouraged to include a comparative analysis to better contextualize the advantages and limitations of their approach. 

Response 9: Thank you for the suggestion. We would like to highlight that our study focuses on a CNN-based approach for coating-thickness monitoring, specifically in the context of unattended sensor nodes performing periodic measurements over long durations, rather than manual inspections or laboratory-based measurements. The sensor node is intended to be deployed inside wind-turbine towers for continuous monitoring. Conventional ultrasonic techniques, such as water-immersion testing, are typically designed for manual operation and short-term measurements, and are not readily applicable for in-situ, long-term monitoring. 

While direct experimental comparisons with traditional methods are therefore not included in this study, our approach is designed to meet the specific requirements of automated, long-term monitoring, which distinguishes it from conventional techniques. In this work, we focus specifically on applying this technique for coating monitoring in real operational conditions considering the coatings’ thicknesses and materials used in this kind of application. 

Comment 10: In summary, while the study presents a promising and innovative application of deep learning in ultrasonic NDT, the manuscript requires substantial revision to address the issues of redundancy, figure organization, technical explanation, and comparative validation. Addressing these concerns will significantly strengthen the paper’s scientific value and readability. 

Response 10: We have notably reduced the length of the Introduction and relocated the necessary theoretical background to the Modelling Approach section as other Reviewer also suggested. This makes the text more concise and improves readability, while retaining only the essential aspects and referring to the relevant literature where appropriate. 

On the other hand, we have improved the figure organization and technical explanation as can be seen in Section 6 (Results and Discussion). 

Regarding the comparative validation, we have to say that our approach operates directly on the measured ultrasound response from the acquisition channel without using external reference signals or prior pulse characterization. In contrast to Wiener deconvolution, which typically requires a known system response to construct an optimal filter, our method processes only the measured signals. Therefore, direct benchmarking against standard deconvolution methods (Wiener, cepstral, AR, wavelet) was not performed in this study, as the primary scope was to demonstrate proof-of-concept and practical applicability of our method within the targeted application scenario.  

We believe that the potential demonstrated by our preliminary experiments constitutes a meaningful scientific contribution for this first publication. We acknowledge the value of systematic comparisons on common datasets using established metrics such as detection rate, ToF error, false positives, ringing indices. Although this can be very complicated, we aim to face this in the future with experimental data in real operation conditions. 

Reviewer 2 Report

Comments and Suggestions for Authors

You’ll find my detailed comments and edits in the attached PDF file

Author Response

  • Comment 1: Generalizability across frequencies. Results are shown only at ~8 MHz. Please validate the full pipeline at 5 MHz and 10 MHz and discuss the resolution–penetration trade-off to substantiate the sub-λ/5 claim in absolute terms.  

Response 1: We fully agree with the reviewer that demonstrating generalizability across frequencies and discussing the resolution–penetration trade-off would be highly interesting and important. However, experimental data obtained from a higher-frequency transducer is not currently available, and therefore this analysis cannot be performed at this stage. 

In the current study, the model has been validated across different material types and coating thicknesses, demonstrating its generalization in these respects. However, the quantitative metrics presented are specific to signals obtained with a transducer of central frequency ~8 MHz, and to synthetic signals generated to replicate these measurements. While the principle of the model is general and could in principle be applied to other frequencies, the achievable resolution and corresponding resolution–penetration characteristics would need to be verified for each case. 

In the revised manuscript, we have clarified this limitation in the abstract, results and conclusion sections (new lines: 29, 763 and 775), specifying that the λ/5 resolution corresponds to signals obtained with a transducer of central frequency ~8 MHz, and have noted that exploration of other frequencies and the resolution–penetration trade-off is planned as future work. 

  • Comment 2: Benchmarking & reproducibility. Add quantitative comparisons against standard deconvolution baselines (e.g., Wiener, cepstral, AR, wavelet) on the same dataset using common metrics (detection rate, ToF error, false positives, ringing index). Report repeatability/reproducibility (same spot, repositioned, inter-operator) under a blinded protocol.   

Response 2: We thank the reviewer for emphasizing the importance of benchmarking and reproducibility.  Our approach operates directly on the measured ultrasound response from the acquisition channel without using external reference signals or prior pulse characterization. In contrast to Wiener deconvolution, which typically requires a known system response to construct an optimal filter, our method processes only the measured signals. 

Direct benchmarking against standard deconvolution methods (Wiener, cepstral, AR, wavelet) was not performed in this study, as the primary scope was to demonstrate proof-of-concept and practical applicability of our method within the targeted application scenario. We believe that the potential demonstrated by our preliminary experiments constitutes a meaningful scientific contribution for this initial publication. We acknowledge the value of systematic comparisons on common datasets using established metrics such as detection rate, ToF error, false positives, ringing indices.  

Implementing multiple baseline models and conducting such comparisons would require substantial additional effort and time. Therefore, we consider this a valuable suggestion, and benchmarking against alternative methods will be addressed as part of future work to further contextualize the advantages of our method. 

Further, we acknowledge the reviewer’s concern regarding repeatability and reproducibility. The measurements used in this study were not all performed by the same operator, and in some cases, measurements were repeated at the same spot at different times. However, at this stage, we are unable to provide quantitative metrics as an analysis for repeatability and reproducibility. Therefore, a blinded protocol, including systematic repositioning and inter-operator evaluation, cannot be implemented under this study. We recognize that a thorough evaluation of repeatability and reproducibility under such controlled conditions is important and plan to explore this in future work. 

  • Comment 3: Absolute thickness & sound speed. Independently measure coating sound speed for each coating type (e.g., pulse-echo on coupons, through-transmission, resonance) and report absolute thickness errors (µm).  

Response 3: We thank the reviewer for this insightful suggestion. Currently, due to limited availability of calibrated coating samples and experimental constraints, we are unable to independently measure the sound speed for each coating type.  

Accurate calibration of speed of sound requires knowing the absolute thickness of the coating on the specific measurement spot from the piezo sensor. To determine absolute thickness over the sensor area, precise measurements of thickness before and after application of coating at the exact same location covering the ultrasonic sensor footprint would be required.  However, since the samples were sourced externally, such controlled pre- and post-application measurements were not feasible within this work's scope.  

Given the variability in coating properties, accurate sound speed measurement requires extensive sample calibration, which is beyond the scope of this initial study. Therefore, we have relied on literature-reported speed of sound value for coating in general, which is a common validated approach in ultrasonic NDT when direct calibration is unavailable.  

Regarding absolute thickness errors, we utilize manufacturer-provided thicknesses for the whole sample as approximate references (referring Table 3), which can include some tolerance in real thickness.  It is important to note that ultrasonic measurements inherently provide an average thickness over the sensor's contact area rather than a point measurement. This averaging effect means that local microscopic variations in coating thickness across the surface are integrated into the measurement, making direct comparison with point-based reference measurements more challenging. 

Having a dedicated methodology for calibration and validation to reliably assess coating thickness estimation accuracy is important. We fully recognize this and plan to focus on developing such procedures in future work. 

We appreciate the reviewer’s understanding and acknowledge these aspects as important directions for future investigations to improve method accuracy and applicability. 

  • Comment 4: Robustness of the deconvolution. Show sensitivity to the regularization parameter K (e.g., L-curve or GCV) and quantify robustness to probe tilt (few degrees), couplant thickness changes, and surface roughness.  

Response 4: We thank the reviewer for this comment regarding the robustness of our deconvolution approach.  

A fully controlled evaluation of robustness—such as sensitivity to the regularization parameter K (e.g., via L-curve or GCV), probe tilt (a few degrees), couplant thickness changes, and surface roughness—was not performed in this study. Extending the assessment to include such systematic analyses will be an important direction for future work. 

However, the real measurements used in this study were performed on different spots of the samples, where we observed that even within the same sample local variations in contact and thickness conditions occur. As a result, the real ultrasound data inherently include variations in coupling conditions and differences in surface roughness depending on the rear-side coatings. These factors represent practical sources of uncertainty that are, to some extent, already reflected in our reported results. Although we do not yet have quantitative measures to present as explicit results, this provides some degree of coverage of variations in measurement conditions. 

  • Comment 5: Synthetic–real domain gap. Discuss differences between k-Wave and experiments (frequency-dependent attenuation/dispersion, non-ideal transducer impulse response) and, if feasible, incorporate these effects in simulation.  

Response 5: It is very challenging to fully bridge the gap between synthetic and real signals, as many aspects of the measurement setup — including the transducer impulse response and experimental noise — as well as non-ideal material properties, cannot be directly incorporated into the simulation. Consequently, while simulation captures the main physics, some differences between synthetic and experimental signals remain unavoidable. We have expanded the discussion of these differences in the manuscript, which can be found at the new line 626: 

“Differences between the simulated and real signals may arise for the following reasons: 

  • Measurement system impulse response and experimental noise. The non-ideal response of the transducer and measurement system cannot be explicitly modeled in k-Wave. In the simulations, the excitation is defined directly as a normal stress with a temporal waveform taken from the first backwall echo of bare steel measurements (one for each steel type, S235 and S355, randomly assigned in the simulations), and spatially modulated with a Gaussian kernel to mimic a focused aperture. Since this reference signal is obtained from pulse-echo measurements with the same transducer used for the coated samples, it is assumed to capture the effect of the transducer’s impulse response. However, it is not strictly representative of the actual transmitted pulse, and thus remains a simplification. Experimental noise, which refers to random fluctuations in the measurement system — e.g., electrical noise, electronic jitter, and environmental vibrations — is also present in the real signals and cannot be incorporated into the simulation. 

  • Frequency-dependent attenuation and dispersion. k-Wave accounts for frequency-dependent attenuation and the corresponding dispersion using a quadratic power-law model (α(f) f²). This corresponds to an ideal thermoviscous loss law, which does not always reflect the actual frequency-dependent absorption of real materials. 

  • Homogeneity and isotropy. The simulations assume constant and isotropic material properties within each layer. This is a good approximation for steel, and although the coatings are intended to be uniform, their application process may lead to small variations in homogeneity that are not captured in the model. 

  • 2D approximation. A 2D geometry is used primarily for computational efficiency, as full 3D simulations at the required resolution would be prohibitively expensive. The 2D assumption is a good approximation when the out-of-plane dimension is much larger than the propagation path, as in our case where the sample thickness is small compared to its lateral area. However, 2D simulations cannot capture out-of-plane scattering, diffraction, or mode conversion, and they confine energy differently than in 3D, which may slightly alter attenuation and energy distribution. 

  • Single-layer coating approximation. In the simulations, the front and rear coatings were modeled as single layers. In contrast, some of the real commercial coatings consist of multiple sublayers. However, the impedance mismatch between these sublayers is assumed to be small, so the dominant contribution to reflections arises from the last sublayer adjacent to air. It is possible that these internal sublayers introduce minor reverberations or phase shifts in the ultrasonic field, but such effects have not been incorporated at this stage. 

This gap between synthetic and experimental signals is difficult to avoid, as it arises from unknown or non-ideal aspects of the measurement setup that are challenging to emulate and incorporate into simulations. In spite of this, the synthetic signals are considered to contain oscillatory patterns sufficiently similar to those in the real signals, allowing the CNN to recognize these patterns and generalize effectively to unseen data.” 

  • Comment 6: Abbreviations (p. 25). Use UT = Ultrasonic Testing (NDT convention). If “ultrasound testbed” is needed, use a distinct acronym (e.g., UTB/UTS).  

Response 6: Thank you for the suggestion. The acronym UTB is now used to refer to the ultrasound testbed. 

  • Comment 7: Units/figures. Alongside “samples,” provide µm conversion using plausible ?3 values (e.g., 2300/2500/2700 m/s) and improve figure legends/contrast.  

Response 7: We have updated the results (figures and tables) to include the micrometer conversion alongside the values in samples, using a reference speed of sound of ?3=2500 m/s. For clarity, the manuscript now includes the following justification in new line: 683: "From now on in the manuscript, results are reported in samples, and for better understanding, the corresponding micrometer values are provided using a reference speed of sound of 2500m/s, so that 1 sample corresponds to approximately 10μm, according to Equation 14." Additionally, the font size of the figure legends in Figures 7, 8, and 9 has been increased to improve readability, and the areas of interest in the subfigures have been zoomed in for easier visual understanding. 

  • Comment 8: Method details. Specify filter passband/coefficients, window type/length, gating bounds, and averaging settings to support reproducibility. 

Response 8: We thank the reviewer for this comment and for the opportunity to provide additional details. We have added more information about bandpass filter design to the manuscript. New line:460-463  

“The bandpass filter is a FIR (Finite Impulse Response) filter designed in MATLAB using the fir1 function with a Hanning window, a filter length of 115, and cutoff frequencies of 2–15 MHz, based on a sampling frequency of 125 MHz. The filter coefficients were quantized for implementation in our custom, non-commercial measurement system, UTB.” 

Echo detection and processing are fully automated; therefore, no gating bounds are applied to select echoes. The process of echo detection has been discussed in our previously published work*. We highlight that the implementation uses our custom hardware, UTB, and some aspects, such as coefficient quantization, are specific to this system. 

*Reference 32 in the manuscript: Thibbotuwa, U.C.; Cortés, A.; Irizar, A. Small ultrasound-based corrosion sensor for intraday corrosion rate estimation. Sensors 2022, 22, 8451. 

Reviewer 3 Report

Comments and Suggestions for Authors

The article is well written and, once minor issues are resolved, it will be suitable for publication in Sensors

In the introduction neural networks are considered AI. It can be true since the hat of AI technique is quite a large hat but it sounds like hype

No errors to highlight but 13 pages of theory seems too much for an article. The authors are advised to summarize as much as possible leaving only essential aspect and relying on the literature references

Table 5 is very informative but in some way, I will try to include the info that the error is 8 microns on your samples. For the reader is not immediately clear what should be the expected range of std.dev ad an example (is that a 0.46 on a max and min of?) Also the median if 0.62 on a range of 0.02 and 2.4? in this case is the distribution skewed? Sorry if this seems trivial questions but I think a better explanation can improve the readability of the paper.

 About that figures that represent the simulated and real signal please consider reporting the difference of the relevant peaks of signal instead juxtaposing them. The simulation is good, but it is also difficult to see that in the interval of interest the signal are “merged”. (it is “good” and “beautiful to see” but not so useful to evaluate i.e. the % error in the simulation if present). Also zooming on the area of interest, commented in the text will be very helpful

About table 6 and also the representation of statistical metrics. I suggest reporting an histogram (the intervals should be the interval of thickness studies) of the errors. In this way it will be possible to understand at glance the reliability of the model (if the error is systematic and remain the same in all the interval it will be natural always to try to investigate thicker samples or if the error is always a percentage of the thickness that would not be true). Also how where the 6 signal (out of 100) chosen? Ther is the possibility of a bias in the sampling and reporting.

Author Response

Comment 1: The article is well written and, once minor issues are resolved, it will be suitable for publication in Sensors 

Response 1: Thank you for this positive comment. 

Comment 2: In the introduction neural networks are considered AI. It can be true since the hat of AI technique is quite a large hat but it sounds like hype.  

Response 2: We agree that the original phrasing was too general. A more accurate description would have been to refer to them as machine learning (ML) and deep learning (DL) models. However, following the reviewers’ suggestions to shorten the introduction, this part was omitted, and the necessary explanations were retained in the state-of-the-art section. 

Comment 3: No errors to highlight but 13 pages of theory seems too much for an article. The authors are advised to summarize as much as possible leaving only essential aspect and relying on the literature references.   

Response 3: In the revised manuscript, we have notably reduced the length of the Introduction and relocated the necessary theoretical background to the Modelling Approach section. This restructuring makes the text more concise and improves readability, while retaining only the essential aspects and referring to the relevant literature where appropriate. 

Comment 4: Table 5 is very informative but in some way, I will try to include the info that the error is 8 microns on your samples. For the reader is not immediately clear what should be the expected range of std.dev ad an example (is that a 0.46 on a max and min of?) Also the median if 0.62 on a range of 0.02 and 2.4? in this case is the distribution skewed? Sorry if this seems trivial questions but I think a better explanation can improve the readability of the paper.   

Response 4: Thank you for pointing this out. We have updated New Table 4 to improve clarity, adding the skewness (0.85) and kurtosis (3.84) values to better characterize the error distribution. The positive skewness indicates that most errors are concentrated toward the lower end of the range, with only a few larger deviations approaching the maximum of 2.4. The kurtosis above three shows that errors are strongly clustered around the central tendency, with extreme values occurring infrequently. Additionally, we have clarified that the standard deviation corresponds to the MAE, and the RMSE, which is used to quantify the model accuracy, has been converted to an approximate thickness error of 8 microns in the table itself. We hope these enhancements make the statistical interpretation of the prediction errors more transparent for the reader. 

Comment 5: About that figures that represent the simulated and real signal please consider reporting the difference of the relevant peaks of signal instead juxtaposing them. The simulation is good, but it is also difficult to see that in the interval of interest the signal are “merged”. (it is “good” and “beautiful to see” but not so useful to evaluate i.e. the % error in the simulation if present). Also zooming on the area of interest, commented in the text will be very helpful.   

Response 5: The subfigures have been revised to provide a zoomed view of the ground truth peaks in the area of interest. Reporting an error between the ground truth and the synthetic reflectivity signals is not meaningful, as the synthetic signals were generated directly from simulations based on that ground truth. Similarly, an error between real and synthetic signals cannot be provided, since there is no one-to-one correspondence: each synthetic signal was generated from a three-layer scenario replicating the experimental conditions, but not from a specific real coated sample measurement. The figures are therefore intended to illustrate the A-scan and corresponding reflectivity signals, highlighting how the synthetic signals reproduce characteristics similar to the real ones for visual comparison. For this purpose, the synthetic examples shown were selected as the closest matches to the real signals in each set, determined by maximizing the cross-correlation. In the revised manuscript, we have also added the similarity metric (derived from the absolute maximum of the cross-correlation) to each subfigure to provide a more quantitative comparison. This metric is explained in the text at the new line 622: “The similarity metric in each subfigure, ranging from 0 to 1 (with values closer to 1 indicating higher similarity), is calculated as the absolute maximum of the cross-correlation between the plotted real and synthetic reflection impulse response signals. For the signals plotted here, this metric ranged between 0.90 and 0.94.”. Moreover, the discussion of potential limitations of the simulations in fully reproducing real signals has also been extended at the new line: 626. 

Comment 6: About table 6 and also the representation of statistical metrics. I suggest reporting an histogram (the intervals should be the interval of thickness studies) of the errors. In this way it will be possible to understand at glance the reliability of the model (if the error is systematic and remain the same in all the interval it will be natural always to try to investigate thicker samples or if the error is always a percentage of the thickness that would not be true). Also how where the 6 signal (out of 100) chosen? There is the possibility of a bias in the sampling and reporting.   

Response 6: Thank you for this suggestion. However, adding a bar plot or histogram of the prediction errors for the real signals is not possible at this stage, since the actual thickness values and the coating’s speed of sound are unknown. We can only rely on the nominal values and a reference speed of sound, which may differ from the real one and therefore does not allow a precise computation of the errors. 

Nevertheless, we found it valuable to perform this analysis on the synthetic signals prediction, where the ground truth is known. In this case, we observed that the prediction error is systematic and remains relatively consistent across different thickness intervals. It can be read in the revised manuscript as (new line: 714): 

“Also, for a clearer visualization of the prediction errors, the synthetic signals were grouped according to their coating thickness into defined intervals, and the mean absolute errors for each interval were plotted in Figure 9, with the number of signals in each interval indicated above the bars. The results show that the errors are consistent across intervals, reflecting a systematic predictive behavior across different thicknesses. Only in the penultimate interval is a slightly higher mean error and standard deviation observed; however, since this analysis is based on a set of 100 synthetic signals with 10–15 signals per interval, this variation lies within the expected range.” 

Regarding the interest in thinner versus thicker coatings, it is indeed true that if the error remains systematic across thickness, thicker coatings are more reliably estimated in terms of relative error. However, as explained in the manuscript, our approach is designed to monitor relative changes in the coating thickness. A systematic error that does not depend on the coating thickness is therefore beneficial for this purpose. Moreover, thinner coatings are of particular interest in some cases due to their protective effect, depending on the material and application, which makes exploring this lower limit valuable. 

Finally, regarding the selection of signals used for plotting, six synthetic signals were chosen as a representative set of different coating thicknesses, and for the real signals, one signal from each set was selected to illustrate the prediction. The overall predictive performance is reported in the tables. 

Round 2

Reviewer 1 Report

Comments and Suggestions for Authors

It can be accepted as it is.

Reviewer 2 Report

Comments and Suggestions for Authors

Thank you for the careful revision. The manuscript is now clearer and technically sound for the intended scope. The following improvements are appreciated:

Clearer description of the deconvolution pipeline and notation; explicit definition of the regularization term K.

Better alignment between synthetic and experimental echoes, with similarity measures reported.

Unit presentation improved (sample-to-µm conversion) and figures/tables are easier to follow.

Abbreviations standardized (UT = Ultrasonic Testing; “testbed” separated), reducing ambiguity.
Overall, the paper reads well and the results are convincingly presented. Only minor editorial polishing remains (e.g., maintain consistent figure legends and keep µm values alongside “samples” throughout).